# Semi-classical quantisation of magnetic solitons in the anisotropic Heisenberg quantum chain

**Yuan Miao[1][*], Enej Ilievski[1,2] and Oleksandr Gamayun[1]**

**1** Institute for Theoretical Physics, University of Amsterdam,
Science Park 904, 1098XH Amsterdam, the Netherlands
**2** Faculty of Mathematics and Physics, University of Ljubljana,
Jadranska 19, 1000 Ljubljana, Slovenia

[*] y.miao@uva.nl

## Abstract

Using the algebro-geometric approach, we study the structure of semi-classical eigenstates in a weakly-anisotropic quantum Heisenberg spin chain. We outline how classical nonlinear spin waves governed by the anisotropic Landau–Lifshitz equation arise as coherent macroscopic low-energy fluctuations of the ferromagnetic ground state. Special emphasis is devoted to the simplest types of solutions, describing precessional motion and elliptic magnetisation waves. The internal magnon structure of classical spin waves is resolved by performing the semi-classical quantisation using the Riemann–Hilbert problem approach. We present an expression for the overlap of two semi-classical eigenstates and discuss how correlation functions at the semi-classical level arise from classical phase-space averaging.



# 1 Introduction

Thermodynamic systems of interacting quantum particles present an outstanding challenge theoretical physics. In spite of their inherent complexity, tremendous progress has been made recently in understanding various facets of quantum many-body systems, including thermalisation [1,2], far-from-equilibrium dynamics [3,4], quantum transport [5,6] and entanglement dynamics [7], especially after the inception of generalised hydrodynamics [8,9]. In this regard, dimensionally constrained models played a prominent role as they not only permit for moderately efficient numerical simulations but also provide a playground for developing and testing analytical approaches. Integrable models are of special interest as they enable us to obtain non-perturbative closed-form solutions that are otherwise rarely available.

This paper is devoted to study the structure and properties of semi-classical eigenstates and the emergence of classical dynamics in the quantum Heisenberg chain, an archetypal example of a quantum many-body system. Emergent integrability in the $\mathcal{N} = 4$ superconformal

Yang–Mills theory [10,11] generated some amount of interest in the semi-classical limits in integrable quantum spin chains. Particularly, the complete semi-classical spectrum of isotropic Heisenberg model has been first described in [12] and subsequently studied in great detail in [13]. The Heisenberg model however appears in a variety of physics applications in the domain of statistical mechanics and is of particular relevance for studying basic principles of out-of-equilibrium many-body dynamics. Following the spirit of Ref. [13], we devote this work to study the semi-classical eigenstates in the easy-axis regime of the anisotropic Heisenberg spin-1/2 chain.

A separate motivation for studying the semi-classical part of the spectrum originates from recent interest in magnetisation transport in integrable quantum chains, where the anisotropic Heisenberg chain plays a prominent role. Several conspicuous similarities with a purely classical magnetisation dynamics governed by the Landau–Lifshitz ferromagnet [14] have been found, both at qualitative and quantitative levels, which firmly point towards a particular type of a classical–quantum correspondence [15–17]. On the classical side, a key piece of evidence rests on an exact solution to the initial value problem with a domain-wall initial profile [16] which discerns three different dynamical regimes depending on the value of anisotropy: (i) ballistic spin transport in the easy-plane regime, (ii) absence of transport in the easy-axis regime and (iii) diffusion with a multiplicative logarithmic correction at the isotropic point. This matches the phenomenology of the quantum isotropic Heisenberg spin-1/2 chain inferred previously in [15]. In this paper we specialise to the easy-axis regime where absence of transport has been linked with the presence of stable kink in the spectrum [16]. Despite that, in what precise manner do such kinks arise as coherent superposition of magnon excitations of the underlying quantum chain remains unknown. The task at hand is to perform semi-classical quantisation of classical nonlinear spin waves that arise in the weakly-anisotropic ferromagnetic Heisenberg chain. The future hope is to study non-equilibrium dynamics directly at the level of semi-classical eigenstates and thus put the classical–quantum correspondence on a firm footing.

It deserves to be emphasised that the aforementioned classical–quantum correspondence of spin transport is not a particularity of the domain wall physics but likewise manifests itself in thermal equilibrium states (i.e. at finite density of magnon excitations). There however it comprises different types of dynamics regimes. Most prominently, in the *isotropic* Heisenberg quantum chain, finite-temperature spin transport (in the zero magnetisation section) is superdiffusive [18–20], belonging to the KPZ universality class (characterised by dynamical exponent $z = 3/2$ [21,22]). Such an anomalous behaviour has been attributed to interacting (and thermally dressed) 'giant magnons', referring to semi-classical eigenstates which at the classical level show up as soliton modes. In contrast, genuine quantum excitations associated to bound magnons states (Bethe strings) become suppressed in this regime [23]. Curiously, this anomalous feature nevertheless entirely disappears upon introducing any amount of interaction anisotropy: on the easy-plane side, one finds ballistic transport characterised by a finite spin Drude weight [24,25], whereas easy-axis anisotropy restores normal diffusion [21,26,27]. In our perspective, these findings offer an extra motivation to carefully examine the structure of semi-classical eigenstates in integrable quantum lattice models.

In this work we use the *asymptotic Bethe ansatz* approach [28] to identify and describe the semi-classical part of the spectrum in the Heisenberg spin-1/2 chain with uniaxial anisotropy. We shall in large part follow the footsteps of Refs. [12,13] by employing an algebro-geometric integration technique [29,30], the Riemann–Hilbert formalism. The main object of interest is the so-called classical spectral curve which provides complete information about the classical finite-gap spectrum of the anisotropic Landau–Lifshitz ferromagnet. Our aim is to make the exposition self-contained and pedagogical. Our attention will be largely devoted to the emergence of special types of semi-classical solutions that represent bions and kinks, for which

anisotropy proves crucial for their stability. In addition, we shall provide closed expression for the semi-classical norms and overlaps, building on earlier works [31–34]. Finally, we briefly discuss the structure of the semi-classical limits of static correlation functions.

The article is structured as follows. In Sec. 2 we begin by briefly introducing the notion of the spectral curve, giving the most basic example of the harmonic oscillator. Next, in Sec. 3 we outline the asymptotic Bethe ansatz technique by applying it to the anisotropic Heisenberg chain. We proceed by solving the classical finite-gap solutions and explicitly working out two specific examples in Sec. 4 and 5. The semi-classical quantisation is then carried out in Sec. 6 and calculations of semi-classical overlaps are given in Sec. 7. Lastly, in Sec. 8 we formulate a conjecture for a classical–quantum correspondence of correlation functions. We finish in Sec. 9 with a conclusion and an outlook.

## 2 Harmonic oscillator: introducing the spectral curve

Prior to delving deep into the realm of many-body systems, we would like to first familiarise the reader with several technical tools that constitute the foundations of the algebro-geometric method to solve differential equations. To this end, we shall describe the semi-classical spectrum of a single quantum-mechanical degree of freedom, the good old quantum harmonic oscillator,

$$\hat{H} = \frac{\hat{p}^2}{2m} + \frac{1}{2}m\omega^2\hat{x}^2. \tag{2.1}$$

As usual, we use a canonical pair of position and momentum operators, satisfying

$$[\hat{x}, \hat{p}] = i\hbar. \tag{2.2}$$

In the language of second quantisation, the Hamiltonian of the quantum harmonic oscillator takes the diagonal form

$$\hat{H} = \hbar\omega\left(\hat{a}^\dagger\hat{a} + \frac{1}{2}\right), \tag{2.3}$$

in terms of the bosonic annihilation operator

$$\hat{a} = \frac{1}{\sqrt{2\hbar}}\left(\sqrt{m\omega}\,\hat{x} + \frac{i}{\sqrt{m\omega}}\,\hat{p}\right). \tag{2.4}$$

The ground state $|0\rangle$ is the Fock vacuum and satisfies $\hat{a}|0\rangle = 0$. Excited eigenstates $|n\rangle$, which obey $\hat{a}^\dagger\hat{a}|n\rangle = n|n\rangle$, are produced by iterative application of the creation operator $\hat{a}^\dagger$ on the ground state $|0\rangle$,

$$|n\rangle = \frac{\left[\hat{a}^\dagger\right]^n}{\sqrt{n!}}|0\rangle, \tag{2.5}$$

such that

$$\hat{H}|n\rangle = E_n|n\rangle = \hbar\omega\left(n + \frac{1}{2}\right)|n\rangle. \tag{2.6}$$

Using a dimensionless coordinate $u = \sqrt{\frac{m\omega}{\hbar}}x$, eigenfunctions $\psi(u) = \langle u|n\rangle$ can be expressed in terms of their normalised logarithmic derivative called *quasi-momentum* (see e.g. [35]),

$$\mathfrak{p}(u) = \frac{\sqrt{\hbar m\omega}}{2}\frac{\partial_u\psi(u)}{\psi(u)}, \tag{2.7}$$

suppressing the subscript $n$ mostly for the sake of clarity. The Schrödinger equation for $\psi(u)$ accordingly transforms into a Ricatti equation

$$\mathfrak{p}^2(u) - i\sqrt{\hbar}\partial_u\mathfrak{p} = 2m(E - V(u)), \qquad V(u) = \frac{\hbar\omega}{2}u^2, \tag{2.8}$$

whereas nodes (i.e. zeros) of the excited wavefunctions $|n\rangle$ have now turned into simple poles. After a short exercise, one can deduce the following representation [35]

$$\mathfrak{p}(u) = i\sqrt{m\hbar\omega}\left(u - \sum_{j=1}^{n}\frac{1}{u-u_j}\right), \tag{2.9}$$

with poles $u_j$ of the quasi-momentum satisfying a simple system of equations

$$u_j = \sum_{k\neq j}\frac{1}{u_j - u_k}. \tag{2.10}$$

In this description, every eigenstate $|n\rangle$ gets assigned a unique set of poles $u_j$, with $j = 1, \ldots, n$ and accordingly Eqs. (2.10) bear a direct analogy to the celebrated Bethe ansatz equations arising in integrable *interacting* quantum systems. By maintaining this analogy, it is furthermore convenient to introduce the $Q$-polynomial

$$Q_n(u) = \prod_{j=1}^{n}(u - u_j), \tag{2.11}$$

which in integrable quantum spin chains corresponds to the Baxter's $Q$-function. By integrating the quasi-momentum we can readily retrieve the wavefunction for the $n$th excited state,

$$\psi_n(u) = \frac{1}{\sqrt{\mathcal{N}}}e^{-u^2/2}Q_n(u), \tag{2.12}$$

with energy $E_n = \left(n + \frac{1}{2}\right)\hbar\omega$ and normalisation $\mathcal{N}$. The solutions to the Bethe-like equations (2.10) are none other than Hermite polynomials $H_n(u)$ of order $n$, that is

$$Q_n(u) = H_n(u). \tag{2.13}$$

Poles $u_j$ are thus identified with zeros (roots) of Hermite polynomials.

## 2.1 Semi-classical limit

We next analyse the semi-classical eigenstates of the quantum harmonic oscillator. Prior to that, we shall briefly remind on some of the well-known results of the classical harmonic oscillator

$$\mathcal{H} = \frac{p^2}{2m} + \frac{1}{2}m\omega^2 x^2. \tag{2.14}$$

Here position $x$ and momentum $p$ are phase-space coordinates with the canonical Poisson bracket

$$\{p, x\} = 1. \tag{2.15}$$

Solutions to the classical harmonic oscillator are conventionally expressed in terms of trigonometric functions

$$p = \sqrt{2mE}\cos(\omega t + \phi), \qquad x = \frac{1}{\omega}\sqrt{\frac{2E}{m}}\sin(\omega t + \phi), \tag{2.16}$$

where $E$ is the value of energy and offset angle $\phi$ is determined by the initial condition.

The classical harmonic oscillator is possibly the simplest example of a dynamical system that is integrable in the Liouville–Arnol'd sense. The associated action-angle pair of variables is simply

$$S = \frac{E}{\omega}, \qquad \varphi = \omega t, \tag{2.17}$$

satisfying the canonical Poisson relation $\{S, \varphi\} = 1$.

**Lax representation.** We proceed by outlining an algebraic reformulation of the above construction. The notion of algebraic integrability rests on the concept of the Lax representation [36]. Here we shortly describe how this construction works by working it out for the toy model – the classical harmonic oscillator. To begin with, we emphasise that the Lax representation for an integrable dynamical system is not unique. In the present case, a suitable choice for the *Lax pair* of $2 \times 2$ matrices $L$ and $M$ is as follows,

$$L(v;t) = \begin{pmatrix} p + im\omega xv & m\omega x - ipv \\ m\omega x - ipv & -p - im\omega xv \end{pmatrix}, \qquad M = \frac{\omega}{2}\begin{pmatrix} 0 & -1 \\ 1 & 0 \end{pmatrix}. \qquad (2.18)$$

Lax matrix $L(v;t)$ depends on time $t$ through $x(t)$ and $p(t)$. In addition, there is *analytic* dependence on the so-called spectral parameter $v \in \mathbb{C}$ which, as we clarify in a moment, is of pivotal importance for algebraic integrability.

The equation of motion for the classical harmonic oscillator is equivalent to the following equation of motion of the Lax matrix $L(v;t)$,

$$\frac{\mathrm{d}}{\mathrm{d}t}L(v;t) = [M, L(v;t)]. \qquad (2.19)$$

Although at first glance it may appear that we have not gained much at all, in the Lax formulation one can immediately recognise the fact that the characteristic polynomial of the Lax matrix,

$$\det(w - L(v)) = 0, \qquad (2.20)$$

is a time independent quantity. Formally speaking, it defines a complex curve $\Sigma(w, v) \subset \mathbb{C}$ called the *spectral curve*. In the current case it is simply an algebraic curve of the form [36]

$$\Sigma: \qquad w^2(v) = 2mE(1 - v^2). \qquad (2.21)$$

Eigenvalues $w_{\pm}(v)$ can be conveniently parametrised in terms of a single double-valued complex function

$$\mathfrak{p}_{\mathrm{cl}}(v) = \sqrt{2mE(1 - v^2)}, \qquad (2.22)$$

called the *classical* quasi-momentum, satisfying $2\cos\mathfrak{p}_{\mathrm{cl}}(v) = \mathrm{tr}(\exp L(v))$. The quasi-momentum $\mathfrak{p}_{\mathrm{cl}}(v)$ features a pair of square-root branch points at $\pm 1$. One can thus interpret the spectral curve as a two-sheet Riemann surface with a branch cut along the interval $\mathcal{I} \equiv [-1, 1]$ and a pole at infinity, $v_{\mathrm{p}} = \infty$. By analyticity, one furthermore has $\oint_{\mathcal{I}} \mathrm{d}\mathfrak{p}_{\mathrm{cl}}(v) = 0$.

**Semi-classical limit.** We now consider the quantum harmonic oscillator and examine the highly-excited eigenstates whose nodes densely distribute along the real axis. Our purpose here is to demonstrate how the classical spectral curve emerges out of a condensate of zeros of the $Q$-polynomial (2.12). This indeed corresponds to the semi-classical limit of large mode numbers $n \to \infty$, with $\hbar \sim 1/n$. From the viewpoint of the classical system, the resulting condensate shows up as a square-root branch cut of the classical spectral curve (2.21). One can likewise think of the reverse process in which the branch cut disintegrates into a collection of simple poles, which is none other than the familiar Bohr–Sommerfeld quantisation rule

$$n \gg 1: \qquad \frac{1}{2\pi}\oint_{\mathcal{C}} \mathrm{d}v\, \mathfrak{p}(v) = \hbar n. \qquad (2.23)$$

The (quantum) quasi-momentum, upon substituting $u = \sqrt{2n}v$ and subsequently taking

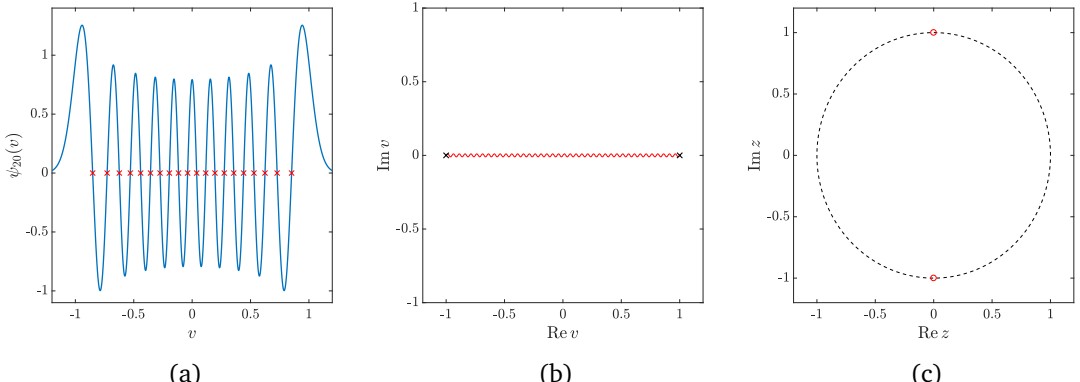

Figure 1: (a) Normalised wavefunction $\psi_{20}(v)$ with mode number 20 and rescaled coordinate $v = u/\sqrt{40}$. Its zeros are marked by red crosses. At large mode numbers, zeros of wavefunctions approach each other and eventually condense. The resulting condensate can be viewed as a square-root branch cut of a spectral curve, cf. Eq. (2.21). Classical spectral curve of the harmonic oscillator in the $v$-plane (b) and $z$-plane (c). The branch cut in (b) gets mapped into two punctures of the Riemann surface. The dashed line indicates the motion of dynamical variable $z^\star$.

the limit $n \to \infty$, becomes

$$
\begin{aligned}
\lim_{n \to \infty} \mathfrak{p}(v) &= \lim_{n \to \infty} i\sqrt{m\hbar\omega} \left[ \sqrt{2n}\, v - \sum_{j=1}^{n} \frac{1}{\sqrt{2n}\, v - u_j} \right] \\
&= i\sqrt{2mn\hbar\omega} \left[ v - \int_{-1}^{1} dy\, \frac{\rho(y)}{v - y} \right],
\end{aligned}
\tag{2.24}
$$

where we have introduced the distribution of zeros of Hermite polynomial

$$
\rho(y) = \lim_{n \to \infty} \frac{1}{n} \sum_{j=1}^{n} \delta\left( y - \frac{u_j}{\sqrt{2n}} \right).
\tag{2.25}
$$

We have thus retrieved the classical quasi-momentum

$$
\mathfrak{p}_{\text{cl}}(v) = i \lim_{n \to \infty} \sqrt{2mn\hbar\omega} \sqrt{v^2 - 1} = \sqrt{2mE(1 - v^2)},
\tag{2.26}
$$

characterised by the classical energy

$$
E = \lim_{n \to \infty} \lim_{\hbar \to E/(\omega n)} n\hbar\omega.
\tag{2.27}
$$

**Canonical angle.** The spectral curve can be associated a dynamical angle-type variable $\varphi$. To find $\varphi$, one requires the solution to the auxiliary linear problem. The latter takes the form of the Lax equations

$$
L\,\psi_{\pm}(v; t) = \lambda_{\pm}(v)\psi_{\pm}(v; t), \qquad \frac{d}{dt}\psi_{\pm}(v; t) = M\,\psi_{\pm}(v; t).
\tag{2.28}
$$

The standard recipe is to identify the canonical angle variables with dynamical poles of the appropriately normalised eigenvectors $\psi_{\pm}$, normally known in the literature as the Baker–Akhiezer vectors [36]. Below we employ a slightly different (but equivalent) approach using

the *squared eigenfunction*, mainly to circumvent the normalisation ambiguity inherent to the Baker–Akhiezer vectors.

Introducing a $2 \times 2$ matrix of eigenvectors $\psi_{\pm}(v, t)$,

$$\boldsymbol{\psi}(v; t) = \big(\psi_{+}(v; t), \psi_{-}(v; t)\big), \tag{2.29}$$

we defined the 'squared eigenfunction' as

$$\boldsymbol{\Psi} = \boldsymbol{\psi}\, \sigma^z\, \boldsymbol{\psi}^{-1}, \tag{2.30}$$

satisfying $\det \boldsymbol{\Psi} = -1$. Notice that $\boldsymbol{\Psi}$ differs from $L$ by the time-independent normalisation. Matrix $\boldsymbol{\Psi}$ itself thus also evolves according to the Lax equation of motion,

$$\frac{\mathrm{d}}{\mathrm{d}t}\boldsymbol{\Psi}(v; t) = [M, \boldsymbol{\Psi}(v; t)]. \tag{2.31}$$

In the present example of the classical harmonic oscillator, the solution to the above differential equation admits the following explicit form

$$\boldsymbol{\Psi}(v; t) = \frac{1}{\sqrt{2mE(1 - v^2)}} \begin{pmatrix} \mathrm{i}m\omega x\left(v - \frac{\mathrm{i}p}{m\omega x}\right) & -\mathrm{i}p\left(v + \frac{\mathrm{i}m\omega x}{p}\right) \\ -\mathrm{i}p\left(v + \frac{\mathrm{i}m\omega x}{p}\right) & -\mathrm{i}m\omega x\left(v - \frac{\mathrm{i}p}{m\omega x}\right) \end{pmatrix}, \tag{2.32}$$

where $x = x(t)$ and $p = p(t)$.

Based on the general rule, the dynamical variables $\gamma_j$ correspond to zeros of the off-diagonal element of the squared eigenfunction $\boldsymbol{\Psi}$. In the language of algebraic geometry, the full set of dynamical variables $\{\gamma_j\}$ constitutes the so-called dynamical divisor of a Riemann surface. Their equations of motion on a surface are governed by a system of differential equations that go under the name of Dubrovin equations. In the case of hyperelliptic algebraic curves of genus $\mathfrak{g}$, the total number of dynamical variables equals $\mathfrak{g} + 1$. Our toy example involves a single degree of freedom and therefore we deal with a surface of genus $\mathfrak{g} = 0$ (i.e. a Riemann sphere). Accordingly, there is a single dynamical variable $\gamma_1 = \gamma_1(t)$, satisfying a simple evolution law

$$\gamma_1(t) = -i \tan \omega t = -\frac{\mathrm{i}p}{m\omega x}. \tag{2.33}$$

To explicitly see this, we perform the following variable transformation,

$$v \mapsto z(v): \qquad v = -\frac{z - 1/z}{z + 1/z}, \tag{2.34}$$

which resolves the square-root type singularities at $v_\star^{\pm}$ and renders $\lambda^2$ a rational function of $z$,

$$\Sigma: \qquad \lambda^2(z) = 2mE\frac{4}{(z + 1/z)^2}. \tag{2.35}$$

The original square-root branch cut with branch points $v_\star^{\pm} = \pm 1$, see Eq. (2.21), have turned into two regular points on the Riemann surface located at $z_\star \in \{0, \infty\}$, whereas the original pole at infinity has two pre-images at two punctures $z_{\mathrm{p}}^{\pm} = \pm i$, one on each Riemann sheet. The dynamical variable $z^*(t) = \exp(\varphi(t))$ satisfies has periodic motion with a linearly-evolving angle variable $\varphi(t) = \omega\, t$, as shown in Fig. 1.

## 2.2 Classical limit of quantum correlation functions

We finally examine the expectation values of quantum observables computed in semi-classical eigenstates. We explain how the latter manifest themselves as classical quantities, thereby establishing an exact classical–quantum correspondence at the level of correlation functions. In the semi-classical limit, the average of a quantum observable $\hat{O}$ should be identified with ergodic phase-space averages

$$\lim_{n\to\infty,\hbar\to\frac{E}{\omega n}} \langle n|\hat{O}(t_1,\cdots,t_m)|n\rangle = \frac{1}{T}\int_0^T \mathrm{d}t\, O(t+t_1,\cdots,t+t_m), \qquad (2.36)$$

where $T = 2\pi/\omega$ denotes the fundamental period.

The correspondence can be readily exemplified on a few particular examples. We first consider, as a simple example, operator $\hat{O} = \hat{x}^2(0)$. Expectation values on a normalised quantum eigenstates read

$$\langle n|\hat{x}^2(0)|n\rangle = \frac{\hbar}{m\omega}\left(n+\frac{1}{2}\right), \qquad (2.37)$$

which in the semi-classical limit yields

$$\lim_{n\to\infty,\hbar\to\frac{E}{\omega n}} \langle n|\hat{x}^2(0)|n\rangle = \frac{E}{m\omega^2} = \frac{\omega}{2\pi}\int_0^{\frac{\omega}{2\pi}} \mathrm{d}t\, x^2(t). \qquad (2.38)$$

The outlined correspondence continues to hold even when $\hat{O}$ consists of several non-commuting operators. To illustrate this, we consider two observables $\hat{O}_1 = \hat{x}(0)\hat{p}(t_0)$ and $\hat{O}_2 = \hat{p}(t_0)\hat{x}(0)$, whose quantum averages yield

$$\langle n|\hat{O}_1|n\rangle = \langle n|\hat{x}(0)\hat{p}(t_0)|n\rangle = \frac{i\hbar}{2}\left[e^{i\omega t_0}(n+1)-e^{-i\omega t_0}n\right], \qquad (2.39)$$

$$\langle n|\hat{O}_2|n\rangle = \langle n|\hat{p}(t_0)\hat{x}(0)|n\rangle = \frac{i\hbar}{2}\left[e^{i\omega t_0}n-e^{-i\omega t_0}(n+1)\right]. \qquad (2.40)$$

Taking again the semi-classical limit, we arrive at

$$\begin{aligned}
\lim_{n\to\infty,\hbar\to\frac{E}{\omega n}} \langle n|\hat{O}_1|n\rangle &= \lim_{n\to\infty,\hbar\to\frac{E}{\omega n}} \langle n|\hat{O}_2|n\rangle \\
&= -\frac{E}{\omega}\sin(\omega t_0) = \frac{\omega}{2\pi}\int_0^{\frac{\omega}{2\pi}} \mathrm{d}t\, x(t)p(t+t_0).
\end{aligned} \qquad (2.41)$$

The upshot of this rather elementary calculation is that expectation values of quantum observables in the semi-classical limit of large mode numbers can be effectively replaced by the corresponding classical observables. There is no ordering ambiguity at the leading order, which only enter in the form of 'quantum corrections', i.e. subleading contributions ($\sim \mathcal{O}(\hbar)$) to the correlation functions.

This concludes our pedagogical introduction to the basic notions of algebro-geometric methods. In the remainder of this article we shall employ the same methodology on a genuine many-body interacting model – the quantum Heisenberg spin-1/2 chain with anisotropic interaction (the XXZ model). Due to multiple degrees of freedom involved, analytical treatment becomes much more challenging and how correlators simplify in the classical limit is no longer a trivial task. Before we come back to this issue in Sec. 8 we first introduce the formalism and other key ingredients.

# 3 Asymptotic Bethe Ansatz: the XXZ Heisenberg model

The anisotropic (XXZ) quantum spin-1/2 chain is often considered as an archetypal model of many-body quantum dynamics. Besides that, it is one of the best studied solvable models by Bethe ansatz [37–39]

$$
\hat{H} = -J \left[ \sum_{j=1}^{L} \hat{S}_j^x \hat{S}_{j+1}^x + \hat{S}_j^y \hat{S}_{j+1}^y + \Delta \left( \hat{S}_j^z \hat{S}_{j+1}^z - \frac{1}{4} \right) \right], \tag{3.1}
$$

using the generators $\hat{S}^\alpha = \hat{\sigma}^\alpha/2$ (where $\hat{\sigma}^\alpha$ are the Pauli matrices) of the $\mathfrak{su}(2)$ spin algebra, satisfying commutation relations $[\hat{\sigma}^\alpha, \hat{\sigma}^\beta] = 2\mathrm{i} \sum_\gamma \epsilon_{\alpha\beta\gamma} \hat{\sigma}^\gamma$. We assume the periodic boundary condition. We shall be interested in the ferromagnetic regime ($J > 0$, $\Delta \geq 0$) and for our convenience set (with no loss of generality) $J = 1$.

There are three phases (regimes) to be distinguished: $\Delta > 1$, $\Delta = 1$ and $0 \leq \Delta < 1$, conventionally called as the gapped, isotropic and gapless regimes, in respective order. This nomenclature refers to properties of the antiferromagnetic ground state. Our focus in this article will be exclusively on the gapped regime. As customary, we parametrise anisotropy as

$$
\Delta = \tfrac{1}{2}(q + q^{-1}) = \cosh\eta, \qquad q = \exp\eta, \qquad \eta \in \mathbb{R}, \tag{3.2}
$$

where $q$ stands for 'deformation parameter' of quantum symmetry algebra $\mathcal{U}_q(\mathfrak{su}(2))$.

The model can be diagonalised by the Bethe Ansatz, a celebrated method invented by Hans Bethe [37]. In what follows, we adopt the ferromagnetic eigenstate $|\uparrow\rangle^{\otimes L}$ as the reference particle pseudovacuum to construct the entire spectrum of eigenstates. In fact, the ferromagnetic states are two-fold degenerate and both ferromagnetic vacua are required to obtain the full spectrum of eigenstates.[1] Since the $z$-component of total spin is conserved, the Hamiltonian block-decomposes into magnetisation sectors labelled by quantum number $M$, pertaining to the number of down-turned spins with respect to pseudovacuum. For fixed $M$, every finite-volume eigenstate, $|\{\vartheta_j\}_{j=1}^M\rangle$, is uniquely characterised by a set of (in general complex-valued) rapidities $\vartheta_j$ called Bethe roots, corresponding to solutions to a coupled system of equations

$$
\left[ \frac{\sin(\vartheta_j + i\frac{\eta}{2})}{\sin(\vartheta_j - i\frac{\eta}{2})} \right]^L \prod_{k \neq j}^{M} \frac{\sin(\vartheta_j - \vartheta_k - i\eta)}{\sin(\vartheta_j - \vartheta_k + i\eta)} = 1, \tag{3.3}
$$

known as the *Bethe equations* (BE). A suggestive physical interpretation underneath the Bethe equations is as follows: the term in the square bracket on the left-hand side equals $e^{\mathrm{i}p}$, with $p = p(\vartheta)$ being the (bare) momentum of a single magnon excitation

$$
p(\vartheta) = -\mathrm{i} \log \frac{\sin(\vartheta + i\frac{\eta}{2})}{\sin(\vartheta - i\frac{\eta}{2})}, \tag{3.4}
$$

whereas the product of quotients of trigonometric functions appearing on the right-hand side is interpreted as a net $U(1)$-valued scattering amplitude acquired by an individual magnon upon undergoing elastic collisions with all the remaining magnons. The total momentum and energy of an eigenstate are obtained by summing over all the constituent magnons, yielding manifestly additive expressions of the form

$$
P(\{\vartheta_j\}_M) = \sum_{j=1}^{M} p(\vartheta_j), \quad E(\{\vartheta_j\}_M) = \sum_{j=1}^{M} \frac{\sin^2 i\eta}{\cos 2\vartheta_j - \cos i\eta}. \tag{3.5}
$$

---

[1]This is no longer the case if the $SU(2)$ invariance is broken by a boundary twist, in which case a single vacuum suffices.

To facilitate the asymptotic analysis of Eqs. (3.3), it is convenient to express them in terms of the Baxter $Q$-functions [40]

$$Q(\vartheta; \{\vartheta_j\}) = \prod_{j=1}^{M} \sin\left(\vartheta - \vartheta_j\right), \tag{3.6}$$

representing 'trigonometric polynomials' of degree $M$ whose zeros (in the fundamental domain) correspond precisely to the Bethe roots of a given eigenstate. Denoting $Q_0(\vartheta) \equiv \sin^L(\vartheta)$, and making use of compact notations for imaginary shifts, $f^{[\pm k]}(\vartheta) \equiv f(\vartheta \pm i\eta/2)$, the Bethe equations can be presented in the form

$$\frac{Q_0^+(\vartheta_j)}{Q_0^-(\vartheta_j)} = \frac{Q_j^{[+2]}(\vartheta_j)}{Q_j^{[-2]}(\vartheta_j)}, \tag{3.7}$$

where $Q_j(\vartheta) \equiv \prod_{k \neq j}^{M} \sin(\vartheta - \vartheta_k)$, or in an equivalent logarithmic form

$$\log Q_0^+(\vartheta_j) - \log Q_0^-(\vartheta_j) = 2n_j \pi i + \log Q_j^{[+2]}(\vartheta_j) - \log Q_j^{[-2]}(\vartheta_j). \tag{3.8}$$

The above form is universal and provides a useful starting point to obtain the semi-classical limits in many other quantum integrable models.

**Thermodynamic limit.** In physics applications one is mostly interested in thermodynamic properties. To this end, one has to infer the structure of solutions to the Bethe equations for large system size, i.e. in the $L \to \infty$ limit with $M \sim \mathcal{O}(L) \to \infty$ magnons. A typical thermodynamic eigenstate comprises of elementary magnon excitations and bound states thereof. The latter show up as compounds of complex Bethe roots (known as the Bethe strings) whose constituent magnon rapidities with unit (imaginary) equidistant separation (neglecting finite-size deviations that are typically exponentially small in $L$)

$$\vartheta_j^{(k)} = \vartheta^{(k)} + \frac{i\eta}{2}(k + 1 - 2j), \qquad j = 1, \ldots, k, \tag{3.9}$$

with the centre $\vartheta^{(k)} \in \mathbb{R}$.

**Low-energy scaling limit.** There exists another thermodynamic limit that is distinct to the one described above. To extract the low-energy spectrum of spin fluctuations at long wavelengths, one has to consider a thermodynamic *scaling* limit by taking both $L$ and $M \sim \mathcal{O}(L)$ large, while additionally demanding that all magnons have low momenta scaling as $\mathcal{O}(1/L)$. This way we are left with a finite number $m \sim \mathcal{O}(1)$ macroscopic bound states. We stress that, in this regime, the Bethe's original argument for the string formation is no longer valid. What happens instead is that bound states become less dense and in general appreciably deformed.

This low-energy eigenstates have been first investigated by Sutherland [41] and afterwards also in Ref. [42], where they are dubbed as 'quantum Bloch walls'. Their classical nature has however been elucidated only later in Ref. [12], establishing an explicit connection with (nonlinear) spin waves governed by the continuous Landau–Lifshitz ferromagnet with isotropic interaction.

The method outlined in Ref. [12] is rather general and can be extended to accommodate also for the interaction anisotropy. As shown below, in the presence of the interaction anisotropy $\eta$, the thermodynamic scaling limit that governs the semi-classical spectrum of low-energy eigenstates requires to additionally assume that anisotropy is weak, $\Delta \gtrapprox 1$. To

be more specific, after reinstating lattice spacing $a$ and writing $\ell \equiv L\,a \in \mathcal{O}(1)$, anisotropy parameter $\eta \sim \mathcal{O}(1/L) \to 0$ has to be rescaled as

$$\eta = \frac{\epsilon\,\ell}{L} + \mathcal{O}\!\left(\frac{1}{L^2}\right), \tag{3.10}$$

with parameter

$$\epsilon \equiv \sqrt{\delta}, \qquad \delta = \frac{2(\Delta-1)}{a^2} \in \mathcal{O}(1), \tag{3.11}$$

kept fixed whilst taking the continuum thermodynamic limit, $L \to \infty$ and $a \to 0$. Here parameter $\ell$ plays the role of a length and later on, in Sec. 3.1, we show that it corresponds precisely to the circumference of the emergent classical phase space.

By first expanding the logarithm of $Q_0^{\pm}$ we find

$$\log Q_0^{\pm}(\vartheta_j) = \log Q_0(\vartheta_j) \pm \frac{i\eta}{2}\frac{\mathrm{d}}{\mathrm{d}\vartheta}\log Q_0(\vartheta)|_{\vartheta=\vartheta_j} - \frac{\eta^2}{8}\frac{\mathrm{d}^2}{\mathrm{d}\vartheta^2}\log Q_0(\vartheta)|_{\vartheta=\vartheta_j} + \mathcal{O}\!\left(\frac{1}{L^2}\right), \tag{3.12}$$

where we have assumed that $\vartheta \in \mathcal{O}(1)$. At this stage it is convenient to perform a change of variable by introducing

$$\mu = \frac{\eta}{\ell}\frac{\mathrm{d}}{\mathrm{d}\vartheta}\log Q_0(\vartheta) = \frac{\epsilon}{\tan\vartheta}, \qquad \mu_j = \lim_{\vartheta \to \vartheta_j}\mu, \tag{3.13}$$

which then yields

$$\log Q_0^+(\vartheta_j) - \log Q_0^-(\vartheta_j) = i\ell\mu_j + \mathcal{O}\!\left(\frac{1}{L^2}\right). \tag{3.14}$$

The part involving $Q(\vartheta)$ can be treated analogously. The details are given in Appendix B. By combining the two contributions, we finally arrive at the following compact representation in the $\mu$-variable [2]

$$\mu_j = \frac{2\pi}{\ell}n_j - \frac{2}{L}\sum_{k \neq j}^{M}\frac{\mu_j\mu_k + \delta}{\mu_j - \mu_k} + \mathcal{O}\!\left(\frac{1}{L}\right). \tag{3.15}$$

Upon taking the thermodynamic scaling limit, the Bethe roots $\mu_j$ condense along certain one-dimensional segments (contours). In general, there are several disjoint contours $\mathcal{C} \equiv \cup_j \mathcal{C}_j$. Accordingly, the Bethe equations turn into singular integral equations of the form

$$\frac{\ell\mu}{2} = \pi n_j - \ell\!\!\fint_{\mathcal{C}}\mathrm{d}\lambda\,\mathcal{K}_{\delta}(\mu,\lambda)\rho(\lambda), \qquad \mu \in \mathcal{C}_j, \tag{3.16}$$

with integral kernel

$$\mathcal{K}_{\delta}(\mu,\lambda) \equiv \frac{\mu\lambda + \delta}{\mu - \lambda}. \tag{3.17}$$

Eqs. (3.16) are known as the *asymptotic Bethe equations* (ABE). To satisfy the reality constraint, the Bethe roots must appear in complex-conjugate pairs, implying that contours $\mathcal{C}_j$ are symmetric under reflection about the real axis. The leading correction term to Eq. (3.15) is of the order $\mathcal{O}(1/L)$ (cf. Appendix B),

$$\frac{1}{L}\pi\rho'(\mu)\ell^2(\mu^2 + \delta)^2\coth\!\left[\pi\ell(\mu^2 + \delta)\rho(\mu)\right]. \tag{3.18}$$

It turns out that this correction to Eqs. (3.16) can only be safely discarded when

$$\pi\ell(\mu^2 + \delta)\rho(\mu) \neq i\pi n, \qquad n \in \mathbb{Z}, \tag{3.19}$$

---

[2]Exact solutions to this algebraic equation for a single mode number can be found in Ref. [43]. A similar construction for the isotropic case appeared in Ref. [44].

which is satisfied when the density of roots near the real axis is sufficiently low. In contrast, when the density of Bethe roots is high enough, the assumptions underlying the above perturbative expansion are no longer justified and one has to resort to the full quantum Bethe equations non-perturbatively (at least in the region near the real axis where the effect is most pronounced). We shall return to this subtlety in Sec. 6.2 and examine it closely on a specific class of solutions. As it turns out, the effect is responsible for emergence of certain special features in the solutions called *condensates* [13].

**Riemann–Hilbert problem.**    The asymptotic Bethe equations (3.16) can be formulated as a Riemann-Hilbert problem. To this end, we define the *spectral resolvent*

$$G(\mu) = \ell \int_{\mathcal{C}} d\lambda \mathcal{K}_{\delta}(\mu, \lambda) \rho(\lambda), \qquad (3.20)$$

and define

$$\mathfrak{p}(\mu) = G(\mu) + \frac{\ell \mu}{2}. \qquad (3.21)$$

At every point $\mu$ along the density contour $\mathcal{C}_j$ function $\mathfrak{p}(\mu)$ experiences a jump discontinuity that is proportional to the density (of Bethe roots) $\rho(\mu)$,

$$\mathfrak{p}(\mu + i0) - \mathfrak{p}(\mu - i0) = 2i\pi\ell(\mu^2 + \delta)\rho(\mu), \qquad \mu \in \mathcal{C}_j, \qquad (3.22)$$

with $\pm i0$ denoting infinitesimal displacements to either side of the contour. [3] Individual contours $\mathcal{C}_j$ can thus be pictured as the $j$th branch cut (of square-root type) of a two-sheeted Riemann surface. The end points of $\mathcal{C}_j$ thus correspond to branch points. In this view, function $\mathfrak{p}(\mu)$ is a double-valued complex function which, apart from contours $\mathcal{C}_j$, is analytic everywhere on the complex $\mu$-plane. A branch cut of square-root type implies that upon crossing it we end up on another Riemann sheet and $\mathfrak{p}(\mu)$ flips its sign. In addition, $\mathfrak{p}(\mu)$ in general picks up an integer multiple of $2\pi$, namely

$$\mathfrak{p}(\mu + i0) + \mathfrak{p}(\mu - i0) = 2\pi n_j, \qquad \mu \in \mathcal{C}_j. \qquad (3.23)$$

Remarkably, $\mathfrak{p}(\mu)$ is precisely the classical quasi-momentum pertaining to the completely integrable classical anisotropic ferromagnet. As explained and demonstrated in the next section, quasi-momentum encodes the eigenvalues of the classical monodromy matrix obtained from a path-ordered exponential of the classical Lax connection.

In the Sec. 6 we shall make a direct comparison between the branch cuts and discrete distributions of Bethe roots for finite system sizes. There we find it useful to rewrite the Riemann-Hilbert problem in terms of variable $\zeta = 1/\mu$. In Appendix A we provide a dictionary between the two parametrisations used.

## 3.1 Anisotropic Landau–Lifshitz field theory

The task at hand is to derive the classical equation of motion for the low-energy spectrum of the Heisenberg XXZ chain. We present how to achieve this in a systematic fashion. The first step is to infer the spatial component of the classical Lax connection from the semi-classical expansion of the quantum monodromy matrix. This will also enable us to explicitly establish that $\mathfrak{p}(\mu)$ from the previous section is truly the classical quasi-momentum that belongs to the classical *axially anisotropic Landau–Lifshitz model*. Written in terms of a $S^2$-valued classical spin field $\vec{\mathcal{S}}(x, t)$, the latter is described by the following partial differential equation

$$\vec{\mathcal{S}}_t = \vec{\mathcal{S}} \times \vec{\mathcal{S}}_{xx} + \vec{\mathcal{S}} \times J\vec{\mathcal{S}}, \qquad J \equiv \text{diag}(0, 0, \delta). \qquad (3.24)$$

---

[3]Orientation of integration contours is in the direction towards the branch point with a negative imaginary part.

Here and subsequently we make use of compact notations $\vec{\mathcal{S}}_t \equiv \partial_t \vec{\mathcal{S}}(x,t)$, $\vec{\mathcal{S}}_x \equiv \partial_x \vec{\mathcal{S}}(x,t)$, and similarly for higher partial derivatives. The Hamiltonian that generates Eq. (3.24) is of the form

$$\mathcal{H} = \frac{1}{2} \int_0^\ell dx \left[ \vec{\mathcal{S}}_x(x) \cdot \vec{\mathcal{S}}_x(x) + \delta - \delta (\mathcal{S}^z(x))^2 \right]. \tag{3.25}$$

**Zero-curvature representation.**  Complete integrability of Eq. (3.24) can be made manifest by recasting it in the form of a *zero-curvature condition*

$$\mathbf{U}_t(\mu; x,t) - \mathbf{V}_x(\mu; x,t) + [\mathbf{U}(\mu; x,t), \mathbf{V}(\mu; x,t)] = 0. \tag{3.26}$$

The latter ensures compatibility of the following *auxiliary linear problem*,

$$\Psi_x(\mu; x,t) = \mathbf{U}(\mu; x,t)\Psi(\mu; x,t), \qquad \Psi_t(\mu; x,t) = \mathbf{V}(\mu; x,t)\Psi(\mu; x,t), \tag{3.27}$$

where [14]

$$\mathbf{U}(\mu; x,t) = \frac{1}{2i} \begin{pmatrix} \mu\,\mathcal{S}^z & \sqrt{\mu^2 + \delta}\,\mathcal{S}^- \\ \sqrt{\mu^2 + \delta}\,\mathcal{S}^+ & -\mu\,\mathcal{S}^z \end{pmatrix}, \tag{3.28}$$

$$\mathbf{V}(\mu; x,t) = \frac{i}{2} \begin{pmatrix} (\mu^2 + \delta)\mathcal{S}^z & \mu\sqrt{\mu^2 + \delta}\,\mathcal{S}^- \\ \mu\sqrt{\mu^2 + \delta}\,\mathcal{S}^+ & -(\mu^2 + \delta)\mathcal{S}^z \end{pmatrix} - \frac{1}{2i} \begin{pmatrix} \mu\,\mathcal{J}_0^z & \sqrt{\mu^2 + \delta}\,\mathcal{J}_0^- \\ \sqrt{\mu^2 + \delta}\,\mathcal{J}_0^+ & -\mu\,\mathcal{J}_0^z \end{pmatrix}, \tag{3.29}$$

are the spatial and the temporal component of the classical Lax connection, respectively, with

$$\mathcal{J}_0 \equiv \vec{\mathcal{S}}_x \times \vec{\mathcal{S}}, \tag{3.30}$$

denoting the spin current density at $\delta = 0$.

**Semi-classical limit.**  We next show how the above (classical) Lax connection can be retrieved from the Lax operator of the quantum chain. The quantum Lax operator is the elementary building block of commuting transfer matrices which facilitate algebraic diagonalisation of the XXZ Hamiltonian (3.1). The fundamental Lax operator $\mathbf{L}(\vartheta)$ acts on a one-site (physical) Hilbert space $\mathcal{V}_p \cong \mathbb{C}^2$ of a spin-1/2 degree of freedom and an auxiliary space $\mathcal{V}_a$ associated to the fundamental representation of the quantum group $\mathcal{U}_q(\mathfrak{su}(2))$ (with deformation parameter $q = e^\eta$), and depends analytically on the complex (spectral) parameter $\vartheta$. It reads explicitly

$$\mathbf{L}(\vartheta) = \frac{1}{\sinh\eta} \begin{pmatrix} \sin(\vartheta + i\eta \mathbf{S}^z) & i\sinh\eta\,\mathbf{S}^- \\ i\sinh\eta\,\mathbf{S}^+ & \sin(\vartheta - i\eta \mathbf{S}^z) \end{pmatrix}, \tag{3.31}$$

in terms of auxiliary spin generators which enclose the $q$-deformed commutation relations

$$[\mathbf{S}^+, \mathbf{S}^-] = [2\mathbf{S}^z]_q, \qquad q^{2\mathbf{S}^z}\mathbf{S}^\pm = q^{\pm 2}\mathbf{S}^\pm q^{2\mathbf{S}^z}, \tag{3.32}$$

using the standard notation $[x]_q = \sinh(\eta x)/\sinh\eta$.

We proceed by constructing the fundamental row transfer matrix of the quantum XXZ spin chain, obtained as a partial trace (over the fundamental auxiliary space $\mathcal{V}_a \cong \mathbb{C}^2$) of the monodromy matrix, i.e. an ordered product of Lax matrices

$$T(\vartheta) = \mathrm{Tr}_{\mathcal{V}_a} \mathbf{M}(\vartheta) = \mathrm{Tr}_{\mathcal{V}_a} \mathbf{L}^{(1)}(\vartheta)\mathbf{L}^{(2)}(\vartheta)\cdots\mathbf{L}^{(N)}(\vartheta). \tag{3.33}$$

Here we have adopted the right-to-left ordering convention and used $\mathbf{L}^{(k)}(\vartheta)$ to denote the embedding into the $k$th factor in an $L$-fold product Hilbert space $\mathcal{H} \cong \mathcal{V}_p^{\otimes L}$. By virtue of the quantum Yang–Baxter relation, matrices $T(\vartheta)$ mutually commute, $[T(\vartheta), T(\vartheta')] = 0$ for

all $\vartheta, \vartheta' \in \mathbb{C}$. Therefore, commuting transfer matrices serve as the generating operator for the local (and nonlocal) conserved charges [12,38]. An infinite tower of commuting fused transfer matrices $T_j(\vartheta)$ with $(j+1)$-dimensional auxiliary unitary representations of $\mathcal{U}_q(\mathfrak{su}(2))$ can be constructed in a similar manner, providing additional quasilocal conservation laws of the XXZ model. While these are of utmost importance for thermodynamic properties at finite energy density (see e.g. Refs. [45–48]), they do not play any role upon taking the semi-classical limit.

We have thus far analysed the asymptotic scaling limit $L \to \infty$ at the level of the Bethe equations by parametrising the interaction parameter as $\eta = \epsilon \ell / L \to 0$ and subsequently taking $\eta \to 0$ ($q \to 1$). Now we do the same at the level of the transfer matrix, where we are allowed to substitute the $q$-deformed spin generators with the fundamental $\mathfrak{su}(2)$ spins, $\mathbf{S}^\alpha \to \hat{S}^\alpha = \frac{1}{2}\hat{\sigma}^\alpha$ for $\alpha \in \{x, y, z\}$. In this limit, the diagonal elements of the quantum Lax operator become

$$\lim_{\eta \to 0} \frac{\sin(\vartheta \mathbb{1} + i\eta \mathbf{S}^z)}{\sinh \eta} = \frac{\sin \vartheta}{\eta} \mathbb{1} + i \cos(\vartheta)\hat{S}^z + \mathcal{O}(\eta), \tag{3.34}$$

whence

$$\mathbf{L}(\vartheta) \simeq \frac{\sin \vartheta}{\eta} \left[ \mathbb{1} + i\eta \begin{pmatrix} \cot(\vartheta)\hat{S}^z & \csc(\vartheta)\hat{S}^- \\ \csc(\vartheta)\hat{S}^+ & -\cot(\vartheta)\hat{S}^z \end{pmatrix} \right]. \tag{3.35}$$

By reinstating the lattice spacing $a = \ell / L$ and using the spectral parameter

$$\mu = \frac{\epsilon}{\tan \vartheta}, \tag{3.36}$$

the Lax matrix writes

$$\mathbf{L}(\mu) \simeq \frac{1}{a\sqrt{\mu^2 + \delta}} \left[ \mathbb{1} + ia \begin{pmatrix} \mu \hat{S}^z & \sqrt{\mu^2 + \delta}\hat{S}^- \\ \sqrt{\mu^2 + \delta}\hat{S}^+ & -\mu \hat{S}^z \end{pmatrix} \right], \tag{3.37}$$

whereas the asymptotic scaling limit of the associated monodromy matrix $\mathbf{M}(\mu)$ is given by the following path-ordered product

$$\frac{\mathbf{M}(\mu)}{(a\sqrt{\mu^2 + \delta})^L} \sim \prod_{j=L}^{1} \left[ \mathbb{1} + ia \begin{pmatrix} \mu \hat{S}_j^z & \sqrt{\mu^2 + \delta}\,\hat{S}_j^- \\ \sqrt{\mu^2 + \delta}\,\hat{S}_j^+ & -\mu \hat{S}_j^z \end{pmatrix} \right]. \tag{3.38}$$

In the final step we replace the quantum spins $\hat{S}^\alpha$ with classical spin variables via $\mathcal{S}_j^\alpha \equiv \mathcal{S}^\alpha(x = j a)$, and subsequently take the continuum limit. We thus arrive as the following semi-classical approximation of the quantum monodromy matrix

$$\mathbf{M}_{\mathrm{cl}}(\mu) \equiv \mathscr{P} \exp \left[ \frac{i}{2} \oint_0^\ell \frac{dx}{2\pi} \mathbf{U}(\mu; x, t) \right], \tag{3.39}$$

with $\mathbf{U}(\mu; x, t)$ being the spatial component of the Lax connection introduced earlier in Eq. (3.28).

## 4 Finite-gap integration method

Having retrieved the classical quasi-momentum $\mathfrak{p}(\mu)$ from the semi-classical expansion of the Heisenberg quantum chain, we proceed to explain how its analytic structure encodes the spectrum of interacting nonlinear phases that characterise classical spin-field configurations. We shall confined our considerations, as usual, to a class of solutions that involve only a finite number of excited nonlinear modes (corresponding to Riemann surfaces of finite genus), commonly known in the literature as the finite-gap solutions [29, 49, 50].

We next describe the full programme for performing an algebro-geometric integration of completely integrable nonlinear partial differential equations which permits to solve the initial value problem. The main steps comprise of:

1. prescribing an appropriately normalised meromorphic differential $d\mathfrak{p}$ on a hyperelliptic Riemann surface of finite genus,

2. constructing the fundamental matrix solution to the associated auxiliary linear problem,

3. identifying the dynamical separated variables and deriving their equations of motion,

4. employing the Abel–Jacobi transformation to obtain canonical action-angle variables satisfying a linear evolution law on a Jacobian hypertorus,

5. expressing physical fields through the solution to the inverse problem.

We stress that the outlined finite-gap integration scheme is not particular to the model at hand but rather applies universality. Moreover, the method has been previously developed also for the Landau–Lifshitz model in Refs. [51,52]. Nevertheless, we wish to offer a different formulation here that we find conceptually simpler. Specifically, we shall avoid the conventional use of the Baker–Akhiezer vectors. Our plan is to first discuss some general aspects and then to provide two explicit realisations for $\mathfrak{g} = 0$ and $\mathfrak{g} = 1$. We do not pay much attention to the construction of the action-angle variables but rather give an explicit time dependence of the physical spin fields in terms of the Riemann theta functions.

## 4.1 Auxiliary linear problem

In this section we describe how to solve the auxiliary linear problem (3.27). By formally integrating along the spatial direction at a fixed time-slice, we have

$$\psi(\mu; x) = \mathscr{P} \exp\left( \int_{x_0}^{x} dx' \, \mathbf{U}(\mu; x') \right) \equiv \mathbf{T}_{\text{cl}}(x, x_0). \tag{4.1}$$

By imposing periodic boundary conditions $\vec{\mathcal{S}}(x + \ell) = \vec{\mathcal{S}}(x)$, we define the monodromy matrix (3.39)

$$\mathbf{M}_{\text{cl}}(\mu) = \mathbf{T}_{\text{cl}}(x_0 + \ell, x_0). \tag{4.2}$$

According to the Bloch theorem, there are two linearly-independent solutions that linearise this translation,

$$\psi(\mu; x + \ell) = \mathbf{\Lambda}(\mu) \psi(\mu; x), \tag{4.3}$$

where Bloch multiplier $\mathbf{\Lambda}(\mu) = \text{diag}(\Lambda_+(\mu), \Lambda_-(\mu))$ is given by eigenvalues of $\mathbf{M}_{\text{cl}}(\mu)$, which (by virtue of the zero-curvature condition (3.26)) are conserved under time evolution, $\partial \Lambda_\pm(\mu)/\partial t = 0$. Since $\text{Tr}\mathbf{U}_{\text{cl}}(\mu) = 0$, monodromy matrix $\mathbf{M}_{\text{cl}}(\mu)$ is unimodular and hence its eigenvalues can be parametrised as $\Lambda_\pm(\mu) = \exp(\pm i \mathfrak{p}(\mu))$ in terms of a single complex-valued function $\mathfrak{p}(\mu)$ called quasi-momentum. Subsequently we make use of the following general parametrisation

$$\mathbf{M}_{\text{cl}}(\mu) = \cos(\mathfrak{p}(\mu)) + i \sin(\mathfrak{p}(\mu))\mathbf{\Psi}(\mu; x_0). \tag{4.4}$$

In analogy with the harmonic oscillator presented in the introductory section 2, we introduce the *squared eigenfunctions* through

$$\mathbf{\Psi}(\mu; x_0) = \psi(\mu; x_0) \sigma^{\text{z}} \psi(\mu; x_0)^{-1}, \tag{4.5}$$

representing a periodic matrix solution (that is $\mathbf{\Psi}(\mu; x + \ell) = \mathbf{\Psi}(\mu; x)$) to the *adjoint* linear system

$$\mathbf{\Psi}_x(\mu; x, t) = \left[ \mathbf{U}(\mu; x, t), \mathbf{\Psi}(\mu; x, t) \right], \qquad \mathbf{\Psi}_t(\mu; x, t) = \left[ \mathbf{V}(\mu; x, t), \mathbf{\Psi}(\mu; x, t) \right]. \tag{4.6}$$

**Uniformisation.** Introducing an uniformised spectral parameter $z$,

$$mu = \frac{1}{2}\left(z - \frac{\delta}{z}\right), \quad \sqrt{\mu^2 + \delta} = \frac{1}{2}\left(z + \frac{\delta}{z}\right), \tag{4.7}$$

the solution to the above linear problem can be sought in the form of a formal Laurent series

$$\Psi(\mu) = \sum_{n=0}^{\infty} \frac{\Psi_n}{z^n}. \tag{4.8}$$

Defining $S^2$-valued matrices

$$\mathbf{S} \equiv \vec{\sigma} \otimes \vec{\mathcal{S}} = \begin{pmatrix} \mathcal{S}^z & \mathcal{S}^- \\ \mathcal{S}^+ & -\mathcal{S}^z \end{pmatrix}, \qquad \widetilde{\mathbf{S}} \equiv \sigma^z \mathbf{S} \sigma^z, \tag{4.9}$$

the first few terms can be written compactly in the form

$$\Psi_0 = \mathbf{S}, \quad \Psi_1 = \mathrm{i}[\mathbf{S}, \mathbf{S}_x], \quad \Psi_2 = -\mathrm{Tr}(\mathbf{S}_x)^2 \mathbf{S} - [\mathbf{S}, [\mathbf{S}, \mathbf{S}_{xx}]] + \frac{\epsilon^2}{4}[\mathbf{S}, [\widetilde{\mathbf{S}}, \mathbf{S}]], \tag{4.10}$$

and so forth.

**Local conserved charges.** The quasi-momentum can be related to the squared eigenfunctions via [14]

$$\mathfrak{p}(z) = -\mathrm{i} \int_0^\ell \mathrm{d}x \, \mathrm{Tr}\big[\mathbf{U}(\Psi + \sigma^z)^{-1}\big] = -\frac{z\,\ell}{4} + \sum_{n=0}^{\infty} \frac{Q_n}{z^n}, \tag{4.11}$$

where coefficients $Q_n$ provide (extensive) local conserved charges for a given solution. The first two take the form (modulo total derivatives)

$$\begin{aligned}
Q_0 &= \frac{\mathrm{i}}{4} \int_0^\ell \mathrm{d}x \, \frac{\mathcal{S}^- \mathcal{S}_x^+ - \mathcal{S}^+ \mathcal{S}_x^-}{1 + \mathcal{S}^z} = -\frac{\mathcal{P}}{2}, \\
Q_1 &= -\frac{1}{2} \int_0^\ell \mathrm{d}x \left[ \vec{\mathcal{S}}_x^2 + \epsilon^2\big(1 - (\mathcal{S}^z)^2\big) - \frac{\epsilon^2}{2} \right] = -\mathcal{H} + \frac{\epsilon^2 \ell}{4},
\end{aligned} \tag{4.12}$$

which yields

$$\mathfrak{p}(\mu) = -\frac{\mu\ell}{2} - \frac{\mathcal{P}}{2} - \frac{\mathcal{H}}{\mu} + \mathcal{O}(\mu^{-2}). \tag{4.13}$$

## 4.2 Finite-gap solutions

The quasi-momentum $\mathfrak{p}(\mu)$ contains only information about the conserved quantities encoded in a particular spectral curve. In order to reconstruct the spin field $\vec{\mathcal{S}}(x,t)$ from the spectral data, we also need knowledge of the dynamical degrees of freedom. Below we outline the algebro-geometric procedure to infer the time evolution of the spin field $\vec{\mathcal{S}}(x,t)$ for the class of *finite-gap solutions*.

To achieve this, we parametrise the squared eigenfunction as

$$\Psi_{\mathfrak{g}}(\mu) = \frac{1}{\sqrt{\mathcal{R}_{2\mathfrak{g}+2}(\mu)}} \begin{pmatrix} a_{\mathfrak{g}+1}(\mu) & \sqrt{\mu^2 + \epsilon^2}\, b_{\mathfrak{g}}(\mu) \\ \sqrt{\mu^2 + \epsilon^2}\, \bar{b}_{\mathfrak{g}}(\mu) & -a_{\mathfrak{g}+1}(\mu) \end{pmatrix}, \tag{4.14}$$

where functions $a_d(\mu)$, $b_d(\mu)$ and $\mathcal{R}_d(\mu)$ are *polynomials* in variable $\mu$ of degree $d$ In particular, $\mathcal{R}_{2\mathfrak{g}+2}(\mu)$ is a polynomial of degree $2\mathfrak{g}+2$,

$$\mathcal{R}_{2\mathfrak{g}+2}(\mu) = \prod_{j=1}^{\mathfrak{g}+1}(\mu - \mu_j)(\mu - \bar{\mu}_j) = \sum_{k=0}^{2\mathfrak{g}+2}(-1)^k r_{2\mathfrak{g}+2-k}\mu^k, \tag{4.15}$$

which specifies a hyperelliptic algebraic curve in $\mathbb{C}^2$,

$$\Sigma: \qquad y^2(\mu) = \mathcal{R}_{2\mathfrak{g}+2}(\mu). \tag{4.16}$$

The curve is fully characterised by $2(\mathfrak{g}+1)$ branch points $\mu_j$ (or, equivalently, symmetric polynomials $r_k$ thereof, with $r_{2\mathfrak{g}+2} = 1$). By the unimodularity constraint, $\det \Psi_{\mathfrak{g}} = -1$, functions $a(\mu)$ and $b(\mu)$ are not independent but are instead subjected to an algebraic relation

$$a_{\mathfrak{g}+1}^2(\mu) + (\mu^2 + \delta) \, b_{\mathfrak{g}}(\mu) \bar{b}_{\mathfrak{g}}(\mu) = \mathcal{R}_{2\mathfrak{g}+2}(\mu). \tag{4.17}$$

From the trace identity (4.11) one can readily infer the following compact expression for the quasi-momentum

$$\mathfrak{p}(\mu) = -\frac{\mu}{2} \int_0^\ell \mathrm{d}x \, \mathcal{S}^z(x) - \frac{\mu^2 + \delta}{4} \int_0^\ell \mathrm{d}x \, \frac{\mathcal{S}^- \bar{b}_{\mathfrak{g}}(\mu) + \mathcal{S}^+ b_{\mathfrak{g}}(\mu)}{\sqrt{\mathcal{R}_{2\mathfrak{g}+2}(\mu)} + a_{\mathfrak{g}+1}(\mu)}. \tag{4.18}$$

This form is compatible with the correct asymptotic expansion about $\mu \to \infty$. A series expansion of $\mathfrak{p}(\mu)$ will thus involve only $(\mathfrak{g}+1)$ functionally independent integrals of motion $Q_n$. They can be expressed as certain functions of coefficients $r_j$ of $\mathcal{R}_{2\mathfrak{g}+2}(\mu)$. Moreover, the total filling fraction $\nu$ with respect to ferromagnetic vacuum $\mathcal{S}^z_{\mathrm{vac}} = 1$,

$$\nu \equiv \frac{1}{2\ell} \int_0^\ell \mathrm{d}x \, (1 - \mathcal{S}^z(x)), \tag{4.19}$$

can be obtained as

$$\mathfrak{p}(\mu = \pm \mathrm{i}\epsilon) = \mp \frac{\mathrm{i}\epsilon}{2} \int_0^\ell \mathrm{d}x \, \mathcal{S}^z(x) = \mp \frac{\mathrm{i}\epsilon\ell}{2}(1 - 2\nu). \tag{4.20}$$

### 4.2.1 Dynamical divisor

Off-diagonal elements of the fundamental matrix solution $\Psi_{\mathfrak{g}}(\mu)$, i.e. dynamical zeros of the $b$-function, provide *dynamical* degrees of freedom of finite-gap solutions [4], enabling the reconstruction of the time-evolved spin field $\vec{\mathcal{S}}(x,t)$.

To satisfy quadratic constraint (4.17), we parametrise

$$b_{\mathfrak{g}}(\mu; x, t) = \mathcal{S}^-(x,t) \prod_{j=1}^{\mathfrak{g}} (\mu - \gamma_j(x,t)), \tag{4.21}$$

where the leading coefficient has been fixed to match the asymptotics at $\mu \to \infty$.

The set $\mathcal{D} = \{\gamma_j\}_{j=1}^{\mathfrak{g}}$ is known as the *dynamical divisor* of the Riemann surface $\Sigma$. Since $\mathcal{S}^-(x,t)$ should also be regarded as an independent dynamical variable, we have in total $(\mathfrak{g}+1)$ dynamical degrees of freedom. This number exactly matches the number of action variables and corresponds to the number of forbidden zones in the finite-gap spectrum.

---

[4]Indeed, function $b_{\mathfrak{g}}(\mu)$ can be interpreted as the classical analogue of the **B**-operator, namely the off-diagonal element of the quantum monodromy matrix $\mathbf{B} \equiv \mathbf{M}_{12}$, whose operator-valued zeros are the quantum separated variables [53].

**Dubrovin equations.** Dynamics of variables $\gamma_j = \gamma_j(x,t)$ takes place on the Riemann surface $\Sigma$. Below we employ an extended dynamical divisor $\mathcal{D}_{\mathrm{ext}}$ by adjoining it two extra non-dynamical variables $\gamma_\pm \equiv \pm i\epsilon$ which we label by $\gamma_{\mathfrak{g}+1}$ and $\gamma_{\mathfrak{g}+2}$, respectively. Using Lagrange interpolation formula, we can then restore functions $a_{\mathfrak{g}+1}(\mu)$ from Eq. (4.17), yielding

$$a_{\mathfrak{g}+1}(\mu;x,t) = \sum_{j=1}^{\mathfrak{g}+2} \sqrt{\mathcal{R}_{2\mathfrak{g}+2}(\gamma_j(x,t))} \prod_{k\neq j}^{\mathfrak{g}+2} \frac{\mu - \gamma_k(x,t)}{\gamma_j(x,t) - \gamma_k(x,t)}. \tag{4.22}$$

The equations of motion (4.6) in terms of $\gamma$-variables from $\mathcal{D}_{\mathrm{ext}}$ take the form

$$\partial_x \gamma_j(x,t) = i\sqrt{\mathcal{R}_{2\mathfrak{g}+2}(\gamma_j(x,t))} \prod_{k\neq j}^{\mathfrak{g}} (\gamma_j(x,t) - \gamma_k(x,t))^{-1},$$

$$\partial_t \gamma_j(x,t) = i\Gamma(x,t) \sqrt{\mathcal{R}_{2\mathfrak{g}+2}(\gamma_j(x,t))} \prod_{k\neq j}^{\mathfrak{g}} (\gamma_j(x,t) - \gamma_k(x,t))^{-1}, \tag{4.23}$$

with

$$\Gamma(x,t) \equiv \frac{r_1}{2} - \sum_{k\neq j}^{\mathfrak{g}} \gamma_k(x,t), \qquad r_1 = \sum_{j=1}^{\mathfrak{g}+1} (\mu_j + \bar{\mu}_j). \tag{4.24}$$

We have obtained a system of differential equations that governs the motion of the dynamical divisor of a Riemann surface, commonly known in the literature under the name of *Dubrovin equations* [29, 49, 54]. The form of these equations is universal, namely they do not depend on the model under consideration. What is model-specific instead are the reconstruction formulae, i.e. how $\gamma$-variables relate to physical fields. A spin field whose target space is a 2-sphere can be described by two degrees of freedom, e.g. the $\mathcal{S}^z$ and $\mathcal{S}^-$ components. These can be restored from Eqs. (4.6), which yields

$$\mathcal{S}^z(x,t) = \sum_{j=1}^{\mathfrak{g}+2} \frac{\sqrt{\mathcal{R}_{2\mathfrak{g}+2}(\gamma_j(x,t))}}{\prod_{k\neq j}^{\mathfrak{g}+2}(\gamma_j(x,t) - \gamma_k(x,t))}, \tag{4.25}$$

and

$$i\frac{\mathcal{S}_x^-(x,t)}{\mathcal{S}^-(x,t)} = \sum_{j=1}^{\mathfrak{g}+2} \frac{\gamma_j \sqrt{\mathcal{R}_{2\mathfrak{g}+2}(\gamma_j(x,t))}}{\prod_{k\neq j}^{\mathfrak{g}+2}(\gamma_j(x,t) - \gamma_k(x,t))}, \tag{4.26}$$

$$i\frac{\mathcal{S}_t^-(x,t)}{\mathcal{S}^-(x,t)} = i\frac{r_1}{2}\frac{\mathcal{S}_x^-(x,t)}{\mathcal{S}^-(x,t)} - \sum_{j=1}^{\mathfrak{g}+2} \frac{\gamma_j\sqrt{\mathcal{R}_{2\mathfrak{g}+2}(\gamma_j)}\left(\sum_{k\neq j}^{\mathfrak{g}+2}\gamma_k(x,t)\right)}{\prod_{k\neq j}^{\mathfrak{g}+2}(\gamma_j(x,t) - \gamma_k(x,t))}. \tag{4.27}$$

**Abel–Jacobi transformation.** Dubrovin equations (4.23) allow for exact integration. To this end, we define the standard basis of $2\mathfrak{g}$ closed cycles on $\Sigma$. They further split into $\mathcal{A}$-cycles and their conjugate $\mathcal{B}$-cycles according to the following prescription: $\mathcal{A}_j$-cycle encircles the the $j$th branch cut $\mathcal{C}_j$ on the upper Riemann sheet of $\Sigma$, whereas $\mathcal{B}_{jk}$ denotes a cycle that passes through cut $\mathcal{C}_j$ on the upper sheet and closes back to itself through $\mathcal{C}_k$, as shown in Fig. 2.

Dubrovin equations for the dynamical divisor on $\Sigma$ can be integrated with aid of the Abel–Jacobi transformation,

$$\varphi_j = 2\pi \sum_{k=1}^{\mathfrak{g}} \int_{\gamma_k(0,0)}^{\gamma_k(x,t)} \omega_j, \tag{4.28}$$

where $\omega_j$ form the basis of holomorphic differentials of the Riemann surface. The above mapping provides a variable transformation from $\gamma$-variables to $\varphi$-variables of the angle-type, $\{\gamma_j(x,t)\} \mapsto \{\varphi_j(x,t)\}$.

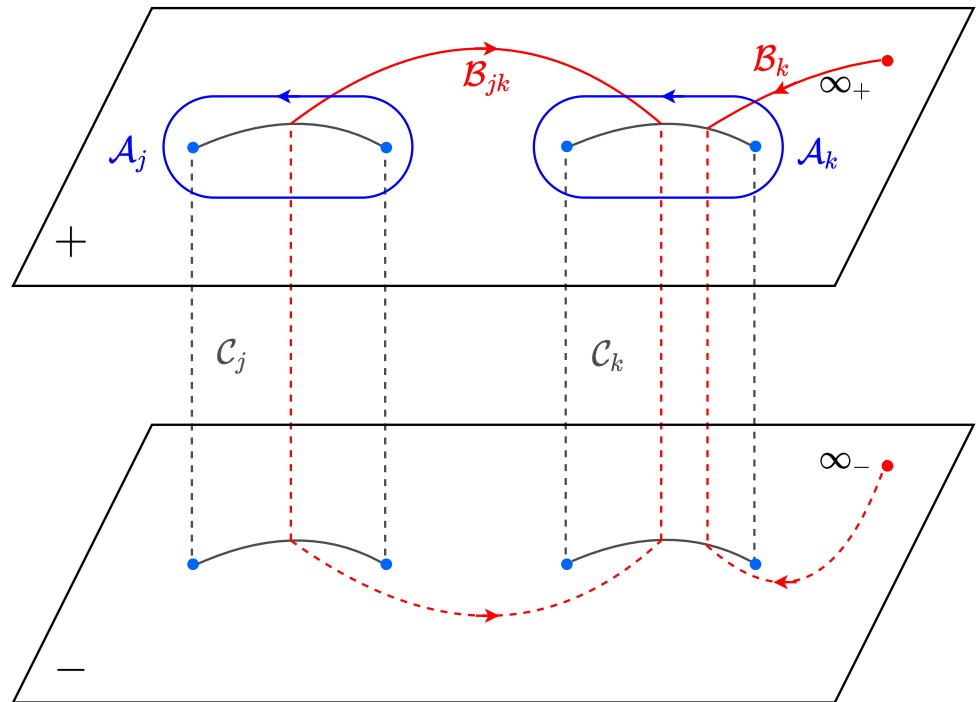

Figure 2: Cycles on a two-sheeted Riemann surface, illustrated on an example of two cuts $\mathcal{C}_j$ and $\mathcal{C}_k$. Upon tunnelling to the other Riemann sheet, the integration orientation is reversed [12].

The holomorphic differentials $\omega_j$ are formally the form

$$\omega_j = \sum_{k=1}^{\mathfrak{g}} \frac{C_{jk}\mu^{\mathfrak{g}-k}\mathrm{d}\mu}{\sqrt{\mathcal{R}_{2\mathfrak{g}+2}(\mu)}}, \qquad j = 1, 2, \ldots \mathfrak{g}. \tag{4.29}$$

Coefficients $C_{jk}$ are determined by requiring canonical normalisation with respect to $\mathcal{A}$-cycles

$$\oint_{\mathcal{A}_j} \omega_k = \delta_{jk}. \tag{4.30}$$

Taking into account Eq. (4.28), equations of motion of the dynamical divisor (4.23) linearise. Indeed, making use of the Lagrange interpolation formulae, one can find

$$\partial_x \varphi(x,t) = 2\pi\mathrm{i}\, C_{j1}, \qquad \partial_t \varphi(x,t) = 2\pi\mathrm{i}\left[\frac{r_1}{2}C_{j1} + C_{j2}\right], \tag{4.31}$$

implying

$$\varphi_j(x,t) = k_j x + w_j t + \varphi_j(0,0), \tag{4.32}$$

with wave numbers $k_j$ are frequencies $w_j$ reading

$$k_j = 2\pi\mathrm{i}C_{j1}, \qquad w_j = 2\pi\mathrm{i}\left[\frac{r_1}{2}C_{j1} + C_{j2}\right]. \tag{4.33}$$

Phases $\varphi_j(x,t)$ satisfy linear evolution in both space and time, exercising a quasiperiodic motion on a Liouville torus $\mathbb{T}^{2\mathfrak{g}}$ of real dimension $2\mathfrak{g}$. For any physically admissible initial condition, $\gamma$-variables evolve along closed trajectories which are homotopically equivalent to $\mathcal{A}$-cycles $\mathcal{A}_j$, such that there is precisely one variable per cycle $\mathcal{C}_j$.

**Periodicity constraints.** A family of periodic (i.e. closed) solutions $\vec{\mathcal{S}}(x) = \vec{\mathcal{S}}(x+\ell)$ is further distinguished by the periodicity of angles, $\varphi_j(x+\ell, t) = \varphi_j(x, t) + 2\pi n_j$ where integers $n_j \in \mathbb{Z}$ specify the mode numbers assigned to each branch cut. Similarly, invariance under translation for a temporal period $T$ implies quantisation of frequencies $w_j$. Under these extra conditions, coefficients $C_{j1}$ and $(r_1/2)C_{j1} + C_{j2}$ become integer-valued, imposing a non-trivial restriction on the admissible algebraic curves.

**Extra degree of freedom.** Now we come to an important subtlety in the above construction: the dynamical separated variables forming the divisor $\mathcal{D} \equiv \{\gamma_j\}_{j=1}^{\mathfrak{g}}$ *do not* provide the complete set of dynamical degrees of freedom for the class of finite-gap solutions. It turns out there is an additional degree of freedom that is not amongst the canonical angle variables (4.28) obtained from dynamical zeros $\gamma_j$ of the $b$-functions. This fact indeed becomes apparent already in the simplest case of genus-zero solutions which, as we in turn demonstrate, governs an evolution of a spin field described by a single angle variable. More generally, for algebraic curves of genus $\mathfrak{g}$, there are thus $(\mathfrak{g}+1)$ phases in total (that is precisely the number of cuts, i.e. half the number of branch points) and accordingly the linearised dynamics takes place in the Liouville torus of real dimension $2(\mathfrak{g}+1)$.

Substituting the form of the $b$-functions (4.21), into Eq. (4.18), and using the restoration formulae (4.25), (4.26), (4.27), we deduce the following identities

$$\int_0^\ell \mathrm{d}x \, \frac{\bar{b}(\mu; x, t)\mathcal{S}^-(x, t)}{\sqrt{\mathcal{R}(\mu)} + a(\mu)} = \int_0^\ell \mathrm{d}x \, \frac{b(\mu; x, t)\mathcal{S}^+(x, t)}{\sqrt{\mathcal{R}(\mu)} + a(\mu)}. \tag{4.34}$$

Introducing an auxiliary function

$$r(\mu, \gamma) = \frac{\sqrt{\mathcal{R}(\mu)} - \sqrt{\mathcal{R}(\gamma)}}{\mu - \gamma} - \frac{1}{2} \sum_{\sigma \in \{+, -\}} \frac{\sqrt{\mathcal{R}(\sigma i\epsilon)} - \sqrt{\mathcal{R}(\gamma)}}{\sigma i\epsilon - \gamma}, \tag{4.35}$$

we can present the quasi-momentum as a period integral of the form

$$\mathfrak{p}(\mu) = -\frac{1}{2} \int_0^\ell \mathrm{d}x \sum_{j=1}^{\mathfrak{g}} \frac{r(\mu, \gamma_j)}{\prod_{k \neq j}(\gamma_j - \gamma_k)}. \tag{4.36}$$

When $\mathfrak{g} = 0$, we put Eqs. (4.21) and (4.22) into Eq. (4.18), which yields

$$\mathfrak{p}(\mu) = -\frac{\ell}{2}\left(\sqrt{\mathcal{R}_2(\mu)} - \frac{1}{2}\left(\sqrt{\mathcal{R}_2(i\epsilon)} + \sqrt{\mathcal{R}_2(-i\epsilon)}\right)\right). \tag{4.37}$$

Comparing Eq. (4.36) with Eq. (4.26), we find that

$$n_{\mathfrak{g}+1} \equiv \frac{1}{2\pi i} \int_0^\ell \mathrm{d}x \, \partial_x \log \mathcal{S}^-, \tag{4.38}$$

provides an extra mode number $n_{\mathfrak{g}+1}$.

Using the fact that $\gamma$-cycles are topologically equivalent to $\mathcal{A}$-cycles that encircle the cuts of $\Sigma$, we can write

$$\mathfrak{p}(\mu) = \frac{i}{2} \sum_{k=1}^{\mathfrak{g}} n_k \oint_{\mathcal{A}_k} \frac{\sqrt{\mathcal{R}(\mu)} - \sqrt{\mathcal{R}(\gamma)}}{\mu - \gamma} \frac{\mathrm{d}\gamma}{\sqrt{\mathcal{R}(\gamma)}} - \pi n_{\mathfrak{g}+1}, \tag{4.39}$$

where $n_k$ is the mode number of the $k$th cut.

On Riemann surface $\Sigma$, quasi-momentum $\mathfrak{p}(\mu)$ is a single-valued function. Alternatively, it can be understood as a double-valued function on the complex spectral plane with $\mu \in \mathbb{C}$,

experiencing jumps on different Riemann sheets upon traversing branch cuts with $\sqrt{\mathcal{R}(\mu)}$ flipping sign. The discontinuity condition for each cut reads

$$\mathfrak{p}(\mu + \mathrm{i}0) + \mathfrak{p}(\mu - \mathrm{i}0) = 2\pi(n_k - n_{\mathfrak{g}+1}), \qquad \mu \in \mathcal{C}_k, \tag{4.40}$$

where infinitesimal shifts $\pm\mathrm{i}0$ pertain to points on the either side of the cut $\mathcal{C}_k$. The above equations are known as the *Riemann–Hilbert problem*. An additional mode number $n_{\mathfrak{g}+1}$ can be inferred from the asymptotic condition at $\mu = \infty_\pm$, reading

$$\mathfrak{p}(\infty_+) + \mathfrak{p}(\infty_-) = -2\pi n_{\mathfrak{g}+1}. \tag{4.41}$$

It proves convenient to introduce a new basis of open $\mathcal{B}$-cycles $\mathcal{B}_k$ (for $k = 1, 2, \ldots, \mathfrak{g} + 1$), connecting infinity $\infty_+$ on the upper sheet with $\infty_-$ on the lower sheet by passing through the cut $\mathcal{C}_k$, as demonstrated in Fig. 2, Eqs. (4.40) alongside (4.41) can be compactly stated as

$$\oint_{\mathcal{B}_k} \mathrm{d}\mathfrak{p}(\mu) = 2\pi n_k, \qquad k = 1, 2, \ldots, \mathfrak{g} + 1. \tag{4.42}$$

**An alternative approach.** Despite Eq. (4.39) provides a solution to the Riemann–Hilbert problem (4.40), the behaviour at $\mu \to \infty$ does not comply with the form of Eq. (4.13). An extra requirement on the spectral curve is needed, namely demanding integrality of the wave numbers (4.33). One route to achieve this is to construct quasi-momentum $\mathfrak{p}(\mu)$ that satisfies the required asymptotics at $\mu \to \infty$ by considering a meromorphic (second-kind) differential

$$\mathrm{d}\mathfrak{p} = -\frac{\ell}{2} \frac{\mathcal{P}_{2\mathfrak{g}+2}(\mu)\mathrm{d}\mu}{\sqrt{\mathcal{R}(\mu)}}, \qquad \mathcal{P}_{2\mathfrak{g}+2}(\mu) = \sum_{j=0}^{\mathfrak{g}+1} c_j \mu^j. \tag{4.43}$$

Then $\mathfrak{p}(\mu)$ is unambiguously determined by demanding analyticity and specifying its asymptotics; the latter readily fixes the highest two coefficients of $\mathcal{P}_{2\mathfrak{g}+2}(\mu)$, namely

$$c_{\mathfrak{g}+1} = 1, \qquad c_{\mathfrak{g}} = -\frac{r_1}{2} = -\frac{1}{2} \sum_{j=1}^{2\mathfrak{g}+2} \mu_j, \tag{4.44}$$

whereas the remaining $m = \mathfrak{g} - 1$ coefficients are fixed by

$$\oint_{\mathcal{A}_j} \mathrm{d}\mathfrak{p}(\mu) = 0, \quad j = 1, 2, \cdots \mathfrak{g}. \tag{4.45}$$

Finally, the spectral curve is uniquely fixed by the condition (4.42).

## 5 Examples of finite-gap solutions

### 5.1 One-cut rational solution

The simplest solutions of the Riemann-Hilbert problem (3.23) belong to algebraic curves of genus zero (Riemann surfaces with a single branch cut). These correspond to quadratic polynomials of the form

$$\mathcal{R}_2(\mu) = (\mu - \mu_1)(\mu - \bar{\mu}_1), \tag{5.1}$$

where branch points $\mu_1, \bar{\mu}_1 \in \mathbb{C}$ are conjugate to one another in order to obey the reality condition. This leads to solutions that involve two real degrees of freedom.

In what follows, we set the classical period to $\ell = 1$. With this choice, the admissible values of the wave numbers are $k = 2\pi n$ with $n$ integer. As a consequence, the branch points cannot be chosen arbitrarily but get "quantised" as well.

Quasi-momentum $\mathfrak{p}(\mu)$ is given by Eq. (4.37). To satisfy the "quantisation condition" and to obtain the prescribed filling fraction (4.20), we have

$$\frac{1}{2}\left(\sqrt{\mathcal{R}_2(i\epsilon)} + \sqrt{\mathcal{R}_2(-i\epsilon)}\right) = 2\pi n, \quad -\frac{1}{4}\left(\sqrt{\mathcal{R}_2(i\epsilon)} - \sqrt{\mathcal{R}_2(-i\epsilon)}\right) = -\frac{i\epsilon}{2}(1 - 2\nu), \quad (5.2)$$

allowing us to parametrise the branch points as

$$\mu_1 + \bar{\mu}_1 = 4\pi n(1 - 2\nu), \quad |\mu_1|^2 = 4\pi^2 n^2 + 4\delta\,\nu(1 - \nu). \quad (5.3)$$

The algebraic curve can be expressed in terms of mode number $n$ and filling fraction $\nu$, reading

$$y^2(\mu) = \mathcal{R}_2(\mu) = \mu^2 - 4\pi n(1 - 2\nu)\mu + 4\pi^2 n^2 + 4\delta(1 - \nu), \quad (5.4)$$

whereas the associated quasi-momentum, cf. Eq. (4.37), can be written as

$$\begin{aligned}
\mathfrak{p}_1(\mu) &= -\pi n - \frac{1}{2}\sqrt{(\mu - \mu_1)(\mu - \bar{\mu}_1)} \\
&= -\pi n - \frac{1}{2}\sqrt{\mu^2 - 4\pi n(1 - 2\nu)\mu + 4\pi^2 n^2 + 4\delta(1 - \nu)}.
\end{aligned} \quad (5.5)$$

On the other hand, the quasi-momentum can also be constructed as an integral of a suitable meromorphic differential on the rational curve (cf. Eq. (4.43)). In $\mathfrak{g} = 0$ case, such differential is described with coefficients $c_0$ and $c_1$,

$$\frac{d\mathfrak{p}_1(\mu)}{d\mu} = -\frac{1}{2}\frac{c_1\mu + c_0}{\sqrt{\mathcal{R}_2(\mu)}}. \quad (5.6)$$

In this case, both coefficients are readily fixed by the asymptotics, see Eq. (4.44), yielding $c_1 = 1$ and $c_0 = -(\mu_1 + \bar{\mu}_1)/2$. By taking an integral of $d\mathfrak{p}_1(\mu)$, we correctly recover the quasi-momentum (5.5).

Notice that presently (i.e. in the zero-genus case) there is no canonical $\mathcal{A}$-cycle. There is a single wave number which can be retrieved by evaluating the 'open' $\mathcal{B}$-cycle

$$k = \oint_{\mathcal{B}_1} d\mathfrak{p}_1 = -(\mathfrak{p}_1(\infty_+) + \mathfrak{p}_1(\infty_-)) = 2\pi n. \quad (5.7)$$

Here $n \in \mathbb{Z}$ is the mode number of the solution.

Coefficients of the asymptotic expansion of $\mathfrak{p}_1$ (cf. Eq. (4.13))

$$\mu \to \infty: \qquad \mathfrak{p}_1(\mu) \sim -\frac{\mu}{2} - \frac{\mathcal{P}}{2} - \frac{\mathcal{H}}{\mu} + \mathcal{O}\left(\mu^{-2}\right), \quad (5.8)$$

provide phase-space averages of local charges evaluated on a particular one-cut solution. The initial two coefficients correspond to total momentum and energy, reading explicitly

$$\mathcal{P} = 2\pi n\nu, \qquad \mathcal{H} = \frac{1}{2}(4\pi^2 n^2 + \delta)\nu(1 - \nu). \quad (5.9)$$

The knowledge of the one-cut quasi-momentum allows to express the dynamics of the spin field $\vec{\mathcal{S}}(x, t)$ in terms of canonical separated $\gamma$-variables. However, since we deal with an algebraic curve of genus zero, we remind that there is no genuine $\gamma$-variable present. Instead,

the only dynamical degree of freedom is the transversal spin component $\mathcal{S}^{-}(x,t)$. We first use Eq. (4.25) to deduce the longitudinal component

$$\mathcal{S}^{z}(x,t) = \frac{\sqrt{\mathcal{R}_2(\gamma_1)}}{(\gamma_1 - \gamma_2)} + \frac{\sqrt{\mathcal{R}_2(\gamma_2)}}{(\gamma_2 - \gamma_1)} = 1 - 2\nu, \tag{5.10}$$

where we have made use of the frozen (i.e. non-dynamical) $\gamma$-variables $\gamma_1 = i\epsilon$ and $\gamma_2 = -i\epsilon$.

In the next step, we can solve Eqs. (4.26) and (4.27) to find the transversal component of the spin field,

$$i\partial_x \log \mathcal{S}^{-} = 2\pi n, \qquad i\left(\partial_t \log \mathcal{S}^{-} - \pi n(1 - 2\nu)\partial_x \log \mathcal{S}^{-}\right) = \delta(1 - 2\nu). \tag{5.11}$$

By imposing normalisation constraint $|\vec{\mathcal{S}}(x,t)| = 1$, we finally arrive at the following general form of the one-cut solution

$$\mathcal{S}^{z}(x,t) = 1 - 2\nu = \cos\theta_0, \qquad \mathcal{S}^{\pm}(x,t) = \sin\theta_0 \exp\left[\pm i(kx - wt)\right], \tag{5.12}$$

where the wave number and frequency are

$$k = 2\pi n, \quad w = (4\pi^2 n^2 + \delta)\cos\theta_0. \tag{5.13}$$

The momentum and energy can be alternatively computed by direct integration using Eq. (4.13)

$$\mathcal{P} = \frac{1}{2i}\int_0^1 dx \frac{\mathcal{S}^{-}\mathcal{S}_x^{+} - \mathcal{S}^{+}\mathcal{S}_x^{-}}{1 + \mathcal{S}^{z}} = 2\pi n\nu,$$

$$\mathcal{H} = \frac{1}{2}\int_0^1 dx \left[\vec{\mathcal{S}}_x \cdot \vec{\mathcal{S}}_x + \delta(1 - (\mathcal{S}^{z})^2)\right] = \frac{1}{2}(4\pi^2 n^2 + \delta)\nu(1 - \nu), \tag{5.14}$$

in agreement with Eq. (5.9).

### 5.1.1 One-cut solution from asymptotic Bethe ansatz

Before describing how to perform semi-classical quantisation of finite-gap solutions (cf. Sec. 6), we wish to briefly discuss the one-cut solutions from the vantage point of low-energy eigenstates in the Heisenberg anisotropic spin chain. Our aim is mainly to identify which solutions to the Bethe equations (3.3) show up classically as one-cut solutions. A numerical method for achieving this is outlined in Appendix E.1. Specifically, we are seeking for low-energy solutions comprising of macroscopically $\mathcal{O}(L)$ excited magnons that condense into a single coherent mode. By fixing the mode number to $n = 1$, we can compute the total momentum $P$ and total energy $E$ of such state with different system sizes $L$ and filling fraction $\nu = M/L$ fixed, see Eqs. (3.5). The results are shown in Fig. 3. On the other hand, we can likewise compute the momenta and energies for classical one-cut solutions, cf. Eq. (5.9), by setting the classical period $\ell = 1$.

Despite that the quantum quasi-momentum in the classical limit reduces to its classical counterpart, we remind that a direct comparison of the respective energies requires to account for an extra rescaling factor of $L$, namely $H \cdot L \sim \mathcal{H}$, where $H$ denotes the eigenvalue of the quantum Hamiltonian defined in Eq. (3.1), while the classical Hamiltonian $\mathcal{H}$ is given by Eq. (3.25).

At the classical level, one-cut solutions describe a simple precessional motion. In some sense, one may view them as a finite-density analogue of Goldstone modes (representing elementary ferromagnetic excitations called magnons). Remarkably, it turns out (see Sec. 6) that quantisation of such solutions can give rise to certain non-perturbative effects (first discussed in Ref. [13]) which necessitate to incorporate quantum corrections into the picture.

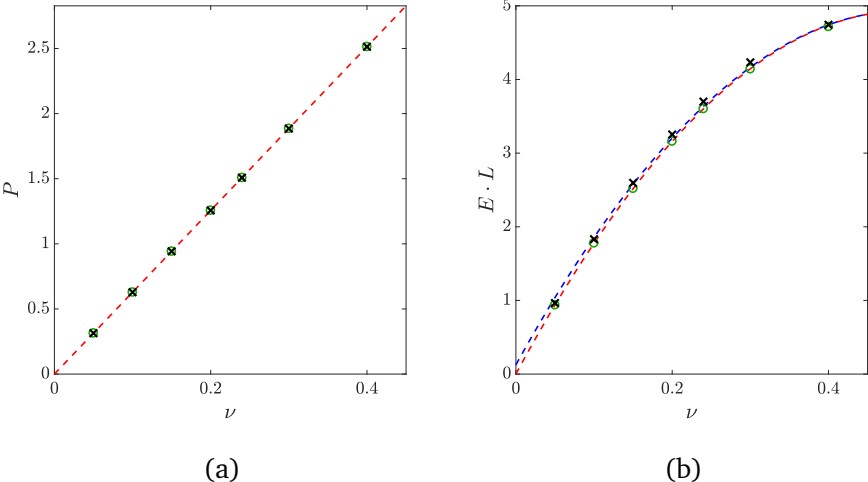

<p style="text-align:center">(a)        (b)</p>

Figure 3: (a) Comparison between momentum of a one-cut solution with mode number $n = 1$ (red dashed line, for both the isotropic and anisotropic interaction $\delta$) and momenta of the corresponding quantum eigenstates (shown for different system sizes $L = M/\nu$, with $M = 30$ and filling fraction $\nu$), with green (black) circles representing the isotropic (anisotropic, with $\delta = 1$) cases. (b) Comparison between energies of a one-cut solution with mode number $n = 1$ (red dashed line for isotropic case and blue dashed line for anisotropic case with $\delta = 1$) and rescaled energies ($E \cdot L$) of the corresponding quantum eigenstates (the same system sizes as in panel (a)).

## 5.2 Bions from degenerate two-cut solutions

As a non-trivial example, we next focus on the class of two-cut solutions. These are, physically speaking, periodic elliptic magnetisation waves which are often referred to as 'cnoidal waves'. The space of two-cut solution is associated to elliptic algebraic curves corresponding to Riemann surfaces of genus $\mathfrak{g} = 1$, characterised by two branch cuts. In the special limit of unit elliptic modulus, the profiles become trigonometric and one retrieves the celebrated soliton solutions. For the particular case of the easy-axis anisotropic Landau–Lifshitz model, there exist special types of two-cut solutions known as *bions*; these represent two-mode bound states formed of a kink and an anti-kink. Moreover, a special degeneration of such a bion solution (upon decompactifying the circumference) produces a static kink. As mentioned in the introduction, kinks are ultimately responsible for the observed freezing of a magnetic domain wall, as demonstrated in [16]. In Sec. 6.3 below, we shall perform the semi-classical quantisation on a bion and study its subtle features. Before that, we derive it here using the outlined integration procedure.

The elliptic curve encoding the bion spectrum has the form

$$\mathcal{R}_{\text{bion}}^2(\mu) = \mathcal{R}_4^2(\mu) = (\mu^2 + \xi_1^2)(\mu^2 + \xi_2^2), \tag{5.15}$$

parametrised by two pairs of complex-conjugate branch points located on the imaginary axis $\text{Re}(\mu) = 0$ at $\mu_j \in \{\pm i\xi_1, \pm i\xi_2\}$, satisfying conditions

$$\xi_1 > \epsilon > 0, \qquad 0 < \xi_2 < \epsilon. \tag{5.16}$$

The upshot here is that the bion solutions can only be found in the regime $\delta > 0$ (the easy-axis regime). In what follows we put, mostly for simplicity, $\delta = \epsilon = 1$. In fact, from the classical equation of motion for the spin field $\vec{\mathcal{S}}(x, t)$ given by Eq. (3.24), the solution at $\delta = 1$ can be

mapped to another solution with $\delta' > 0$ by a simple rescaling

$$\vec{\mathcal{S}}(x,t) \quad \mapsto \quad \vec{\mathcal{S}}(x' = \epsilon\, x, t' = \delta\, t). \tag{5.17}$$

We next outline how to reconstruct of the spin field from the algebraic curve. To set the stage, we introduce the standard full elliptic integrals, cf. Appendix C,

$$K_1 = K\left(\frac{\xi_2^2}{\xi_1^2}\right), \qquad K_2 = K\left(1 - \frac{\xi_2^2}{\xi_1^2}\right). \tag{5.18}$$

When the argument of the elliptic function is omitted we adopt that $k = \xi_2/\xi_1$.

Since the Riemann surface involves two branch cuts, this time we do have a genuine dynamical $\gamma$-variable $\gamma_1(x,t)$. As said earlier, there exist two extra non-dynamical variables pinned to locations $\gamma_2 = i$ and $\gamma_3 = -i$ (recall that $\epsilon = 1$). From the Dubrovin equations (4.23) we find

$$\gamma_1(x,t) = \frac{i\xi_1}{\mathrm{sn}(u)} = -i\xi_2\,\mathrm{sn}(u + iK_2), \qquad u = \xi_1 x + \varrho, \tag{5.19}$$

where we have used $\mathrm{sn}(x + iK_2)\mathrm{sn}(x)k = 1$, and used $\varrho$ to denote the integration constant. Remarkably, it turns out that in this particular subclass of two-cut solutions even $\gamma_1$ is *static*.

Using the reconstruction formulae (4.25) for $\mathcal{S}^z$ component, we thus have

$$\mathcal{S}^z = \frac{\sqrt{\mathcal{R}_4(\gamma_1)} + i\sqrt{\mathcal{R}_4(i)}\gamma_1}{\gamma_1^2 + 1}. \tag{5.20}$$

To fix the integration constant $\varrho$, we impose the reality condition $\mathcal{S}^z \in \mathbb{R}$,

$$\varrho = \mathrm{arcsn}\left(i\frac{\xi_1}{\xi_2}\sqrt{\frac{1 - \xi_2^2}{\xi_1^2 - 1}}\right), \tag{5.21}$$

which yields a time-independent profile

$$\mathcal{S}^z(x) = -\xi_2\frac{\mathrm{cn}(\xi_1 x)}{\mathrm{dn}(\xi_1 x)}. \tag{5.22}$$

The solution has a spatial period

$$\ell = \frac{4nK_1}{\xi_1}, \tag{5.23}$$

where $\pm n$ are the mode numbers associated with the two cuts. Variable $\gamma_1$ can be therefore expressed as

$$\gamma_1(x) = -\xi_1\xi_2\frac{-i\xi_1\sqrt{\mathcal{R}_4(i)}\,\mathrm{cn}(\xi_1 x)\mathrm{dn}(\xi_1 x) + i(\xi_1^2 - \xi_2^2)\mathrm{sn}(\xi_1 x)}{\xi_1^2(1 - \xi_2^2) + (\xi_1^2 - 1)\xi_2^2\,\mathrm{sn}(\xi_1 x)^2}. \tag{5.24}$$

The other independent component of the spin field, say $\mathcal{S}^-(x,t)$, can be reconstructed with aid of formulae (4.26) and (4.27), i.e.

$$\mathcal{S}^-(x,t) = \frac{1}{\sqrt{1 + \gamma_1^2(x,t)}}\exp\left(-i\int dx\,\frac{-i\sqrt{\mathcal{R}_4(i)}}{1 + \gamma_1^2(x,t)}\right). \tag{5.25}$$

Using the properties of elliptic functions, we have

$$\mathcal{S}^-(x) = C\frac{\Theta(\xi_1 x + \varrho + iK_2)}{\Theta(\xi_1 x + K_1)}\exp\left(\xi_1 x\frac{\pi i}{2K_1} + \xi x\frac{\Theta'(\beta)}{\Theta(\beta)}\right), \tag{5.26}$$

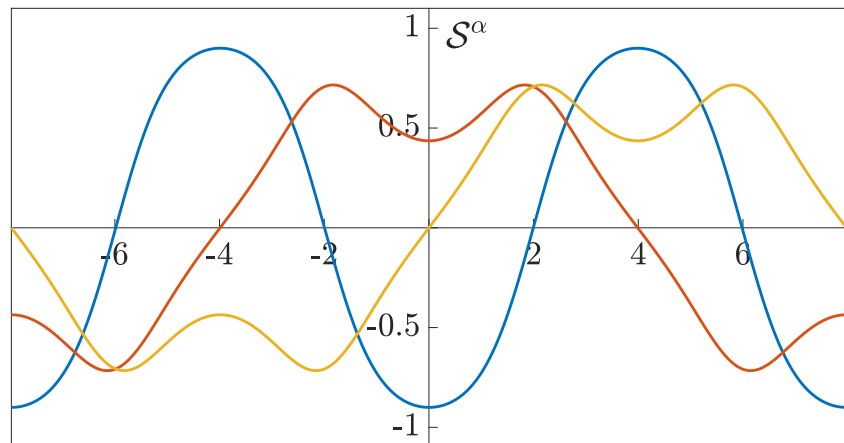

Figure 4: Spin-field components of a typical bion solution, depicted for anisotropy parameter $\delta = 1$ and branch points $\xi_1 \simeq 1.0583559$ and $\xi_2 = 0.9$. Components $S^z(x)$, $S^x(x) = \operatorname{Re} S^-(x)$, and $S^y(x) = -\operatorname{Im} S^-(x)$ are shown by blue, yellow and red curves, respectively. The branch points are determined using the method in Appendix E. The bion solution is periodic with period $\ell_2 = 4K_1/\xi_1 \simeq 15.956517$.

where an auxiliary function $\Theta(x)$ is defined as (cf. Appendix C)

$$\Theta(x) = \vartheta_3\left(\frac{\pi x}{2K_1} + \frac{\pi}{2}, -i\frac{K_2}{K_1}\right). \tag{5.27}$$

Finally, constant $C$ is can be fixed by requiring normalisation $\vec{S}^2 = 1$, yielding

$$S^-(x) = \sqrt{1-\xi_2^2}\frac{\Theta(\xi_1 x + \rho + iK_2)}{\Theta(\xi_1 x + K_1)}\exp\left(\xi_1 x \frac{\pi i}{2K_1} + \xi x \frac{\Theta'(\beta)}{\Theta(\beta)} + i\phi_0\right), \tag{5.28}$$

where $\phi_0 \in \mathbb{R}$ is a phase that is determined by the initial condition. We just found that for the bion solution $S^-$ is time-independent as well. A representative example of a bion solution is shown in Fig. 4.

**Static kink.** As our final example, we consider a particular degeneration of a bion solution which produces a kink. One can think of it as 'soliton limit' which generally corresponds to 'blowing up' the period $\ell$. This requires to bring both branch points of $\sqrt{\mathcal{R}_4(\mu)}$ in the upper-half plane together to meet at $i\epsilon$, that is sending $\xi_{1,2} \to \epsilon$ (presently $\epsilon = 1$) and thus collapsing both branch cuts to a point. In order to perform this soliton limit, we first shift the argument of the elliptic function by quarter period $K_1/\xi_1$; for instance the $S^z$ field takes the form (cf. Eqs. (C.6))

$$S^z(x) = -\xi_1\frac{\operatorname{cn}(\xi_1(x + K_1/\xi_1))}{\operatorname{dn}(\xi_1(x + K_1/\xi_1))} = \xi_1 \operatorname{sn}(\xi_1 x). \tag{5.29}$$

Now taking the limits $\xi_{1,2} \to 1$ and accordingly $k = \xi_2/\xi_1 \to 1$, we find

$$S^z_{\mathrm{kink}}(x) = \tanh(x), \tag{5.30}$$

which is none other than a *static kink*! The transversal components can be easily deduced from the equation of motion (3.24), yielding

$$S^-_{\mathrm{kink}}(x) = \operatorname{sech}(x)\,e^{i\phi_0}, \tag{5.31}$$

where $\phi_0 \in \mathbb{R}$ is a phase determined by the initial condition.

# 6  Semi-classical quantisation

We have now fully prepared the stage to carry out the semi-classical quantisation on finite-gap solutions. In this respect, the associated spectral curves (encoding complete information about the conserved quantities) will be of vital importance.

An important remark is in order at this point. First, recall that moduli of algebraic curves are completely fixed by locations of the square-root branch points, i.e. roots of polynomial $\mathcal{R}_{2\mathfrak{g}+2}(\mu)$, which (by the reality condition) must always appear in complex-conjugate pairs. Local conserved charges are expressible as functions of symmetric polynomials of branch points, namely coefficients $r_j$ of $\mathcal{R}_{2\mathfrak{g}+2}(\mu)$. While branch points $\{\mu_j\}$ themselves are directly linked physical quantities, the branch cuts (of square-root type) obtained by pairwise connecting the branch points are not physical but merely a matter of convention. There is indeed plenty of freedom in assigning branch cuts to a given set of branch points. For instance, the prevalent choice in the finite-gap literature [36] is to place the cuts along straight vertical lines connecting every complex conjugate pair of branch points, which are in turn encircled by the canonical basis $\mathcal{A}$-cycles. Such a choice is, purely from the standpoint of classical finite-gap solutions, perfectly adequate. One the other hand, if classical solutions are instead thought of as emergent macroscopic bound states of magnons of the underlying quantum spin chain, it is natural to cut the surface along one-dimensional disjoint segments associated to forbidden zones of the classical transfer function [12], corresponding to contours on which magnons (Bethe roots) condense. This prescription for the branch cuts is physically distinguished. As demonstrated in turn, such physical cuts not only appreciably differ from the conventional straight cuts in general, but also undergo the phenomenon of condensate formation. Computing the Bethe root densities along the physical contours is thus the essential step to perform the semi-classical quantisation.

## 6.1  Physical contours

In this section we describe a general procedure to determine the physical contours. The algorithm we employ has been proposed and implemented in Ref. [13]. We shall also facilitate a direct comparison with exact low-momentum quantum eigenstates at finite $L$ in the weakly-anisotropic regime. For our convenience, we carry out this computation in $\zeta$-plane [5] by applying the following anti-holomorphic transformation [6]

$$\mu \mapsto \zeta(\mu): \qquad \zeta = \frac{1}{\mu} = \frac{\tan\vartheta}{\epsilon}. \tag{6.1}$$

Firstly, we introduce the *complex* density function

$$\rho(\zeta) = \frac{\mathfrak{p}(\zeta + \mathrm{i}0) - \mathfrak{p}(\zeta - \mathrm{i}0)}{2\mathrm{i}\pi\ell(1 + \delta\zeta^2)} = \pm\frac{\mathfrak{p}(\zeta \pm \mathrm{i}0) - \pi n_j}{\mathrm{i}\pi\ell(1 + \delta\zeta^2)}, \qquad \zeta \in \mathcal{C}_j, \tag{6.2}$$

where $n_j \in \mathbb{Z}$ is the mode number associated to cut $\mathcal{C}_j$. By virtue of the second equality in Eqs. (6.2), the density function $\rho(\zeta)$ can be defined on the entire Riemann surface. As a direct consequence of $\mathfrak{p}(\zeta = \zeta_\star) = \pi n_j$ at the square root branch points $\zeta_\star \in \{\zeta_j, \bar{\zeta}_j\}$, the density always vanishes, $\rho(\zeta_\star) = 0$. In a small neighbourhood around it, one finds $\rho(\zeta = \zeta_\star + \varepsilon) = \mathcal{O}(\sqrt{\varepsilon})$. Away from the branch points $\rho(\zeta)$ in general takes complex values.

---

[5]Our variable $\zeta$ is analogous to variable $x$ used in Refs. [12, 13, 55] in the case of the isotropic Heisenberg model.

[6]Upon this transformation, orientation of integration contours gets reversed.

The task is to determine the physical contours $\mathcal{C}_j$ The latter can be singled out by the following condition: starting from the branch point $\zeta_j$, the integrated density differential $\rho(\zeta)\mathrm{d}\zeta$ must always remain real,

$$\int_{\zeta_j}^{\zeta}\mathrm{d}\zeta'\rho(\zeta')\in\mathbb{R}\qquad\text{for}\qquad\zeta\in\mathcal{C}_j. \tag{6.3}$$

This prescription has a transparent physical interpretation; physically $\rho(\zeta)\mathrm{d}\zeta$ corresponds to the number of excitations (Bethe roots) within an infinitesimal interval in $\zeta$-plane, which is a positive definite quantity by definition. This condition alone is however not sufficient yet. In particular, it turns out that there are three distinct contours emanating out of each branch point compatible with the above positivity requirement [13]. Amongst those, one of the contour carries an infinite filling fraction and can be thus immediately ruled out as unphysical. Out of the remaining two contours, only one can be physical. The defining condition is that the total filling fraction does not exceed the threshold value of maximal total filling $\nu_{\max}=1/2$, that is

$$\int_{\mathcal{C}}\mathrm{d}\zeta'\rho(\zeta')\le\nu_{\max},\qquad\mathcal{C}=\bigcup_j\mathcal{C}_j. \tag{6.4}$$

Consider now a certain reference finite-gap solution. To quantise it at the semi-classical level, every density contour (physical cut) has to be dissolved into a large (but finite) number of individual magnons. This invariably requires to reintroduce the length $L$ of the underlying spin chain, thus rendering the total magnetisation carried by individual coherent states to come in integer quanta of $M_j\sim\mathcal{O}(L)$. The precise requirements are that (i) $M_j/L\approx\nu_j$ and (ii) that the Bethe roots are distributed approximately with density $\rho_j(\zeta)$ along $\mathcal{C}_j$. Here it is important to make a distinction with the exact quantisation which instead takes quantum fluctuations fully into account to all orders in the effective Planck constant. This means, in other words, that the semi-classical solutions produced with the outlined procedure can be at best an approximation of finite-volume exact quantum-mechanical eigenstates at large wavelengths, while a full non-perturbative (i.e. exact) quantisation would require solving the Bethe ansatz equations (3.3).

**Single contour at low density.** To benchmark the above procedure, we proceed by illustrating first how one-cut solutions emerge as semi-classical eigenstates in the anisotropic gapped Heisenberg spin-1/2 chain.

We shall first suppose that the filling fraction of a physical cut is sufficiently *low*, ensuring that the finite-size effects (cf. Eqs. (3.18) and (B.16)) can be safely neglected at large system sizes. We then find that the Bethe roots patterns which solve the asymptotic Bethe equations distribute along certain arcs in the complex rapidity plane, as exemplified in Fig. 5. To be concrete, we put anisotropy to $\delta=1$ and set the filling fraction to $\nu=0.1$ and the mode number to $n=1$. Using the above prescription, we next compute the density contour satisfying Eqs. (6.3) and (6.4), as shown in Fig. 5 (the procedure to numerically solve the Bethe ansatz equations (3.3) for the case of a single quantised one-cut solution is described in Appendix E.1). Upon taking the $L\to\infty$ limit and rescaling the rapidity variable, the semi-classical eigenstate will eventually be described by a dense arrangement of Bethe roots distributing along the contour specified by the conditions (6.3) and (6.4).

By ramping up the filling fraction $\nu$, we observe that 'quantum fluctuations' (contained in higher order terms in the ABE) progressive amplify. As announced earlier, this eventually leads to a critical phenomenon of condensate formation. This feature will be closely examined in the next section.

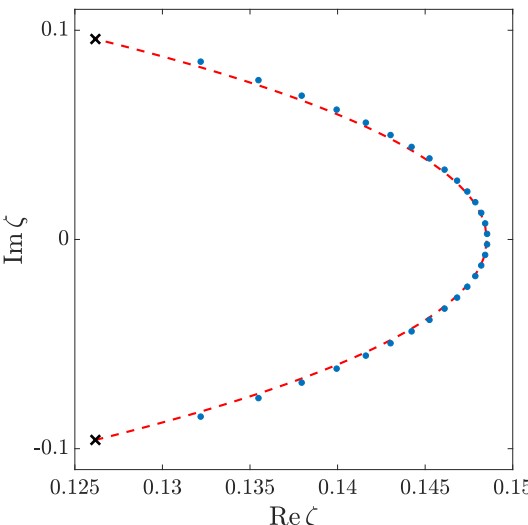

Figure 5: Direct comparison between the physical density contour of a classical one-cut solution (red dashed line, corresponding to anisotropy $\delta = 1$, filling fraction $\nu = 0.1$ and mode number $n = 1$) determined by imposing the positivity condition (6.3), and the corresponding solution to Bethe ansatz equations (3.3) with $M = 30$ Bethe roots, system size $L = 300$ and anisotropy $\eta = \sqrt{\delta}/L = 1/300$ (blue dots). The Bethe roots are given by $\zeta_j = \tan(\vartheta_j)/\sqrt{\delta}$.

## 6.2   Formation of condensates

*Condensates* refer to segments of a uniform density as a part of a physical contour. We borrowed this name from Refs. [12, 56] where (to the best of our knowledge) such objects have been first identified. Condensates enter the picture whenever the maximal density along one of the physical contours exceeds a particular critical value which is signalled by a divergence of the finite-size correction given by Eq. (3.18) (see also Eq. (B.16)).

We shall first examine the phenomenon on the simplest case of the one-cut solution, using the $\zeta$-plane parametrisation. By starting at some sufficiently low filling fraction $\nu$ we can observed that upon gradually increasing the filling fraction, the value of $\rho(\zeta)(1 + \delta\zeta^2)$ on real axis approaches the value of i. This value is reached at the critical filling fraction $\nu_{\text{crit}}$, precisely when quantum fluctuation of order $\mathcal{O}(1/L)$ become divergent, as indicated by Eq. (B.16). For larger fillings $\nu > \nu_{\text{crit}}$, the density contour develops a vertically straight segment of unit uniform density. Such a condensate appears first on the real axis (right after crossing $\nu_{\text{cric}}$) and progressively expands outwards upon further increasing $\nu$. From the viewpoint of the underlying quantum chain, the spacing between constituent Bethe roots is always equal to i$\eta$. One can therefore think of condensates as giant regular Bethe strings. In the complex spectral plane associated to finite-gap solutions, the ends points of a condensate correspond to branch cuts of *logarithmic* type. [7]

In spite of appearance of an additional condensate above $\nu > \nu_{\text{crit}}$, it is still possible to extract the physical contour solely from the knowledge of a finite-gap solution by taking into account conditions (6.3) and (6.4).

Emergence of condensates is intimately tied to the notion of 'fluctuation points', playing

---

[7]Logarithmic branch cuts get likewise produced in a well-known soliton degeneration process, corresponding to merging two nearby standard square-root branch cuts by coalescing their type branch points in a pairwise manner. In effect, finite-gap quasi-momentum is no longer meromorphic. Condensates are different in this respect, as their quasi-momentum differential remains meromorphic all the way through.

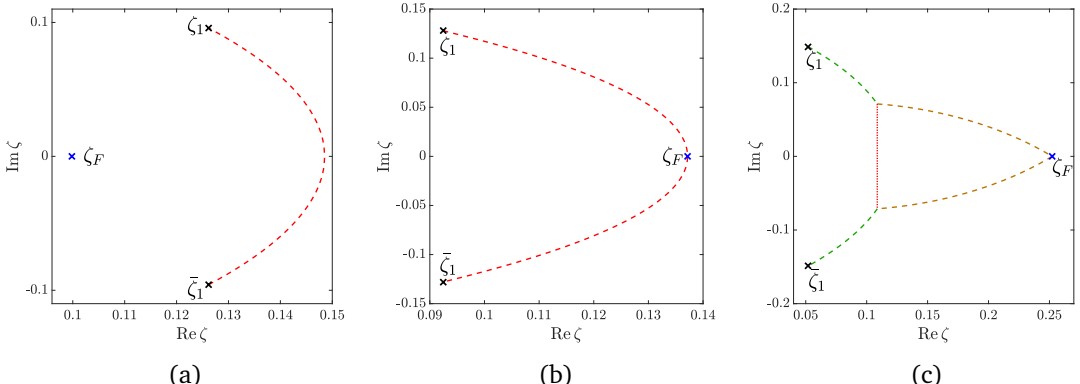

Figure 6: Branch points ($\zeta_1$ and $\bar{\zeta}_1$) and fluctuation points ($\zeta_F$) for the one-cut solution with anisotropy parameter $\delta = 1$ and various filling fractions $\nu = \{0.1, 0.206354963, 1/3\}$, increasing from left to right. Panel (b) corresponds to the critical filling fraction $\nu_c \simeq 0.206354963$, when the fluctuation point $\zeta_F$ collide with the physical cut.

a pivotal role in classical modulation stability theory [57]. Fluctuation points can perceived as small fluctuations of a reference finite-gap solution, corresponding to tiny cuts that possess infinitesimal filling fractions. Upon increasing their filling fraction, they grow up into nonlinear finite-gap mode. Let us slightly expand on this point. Imagine a reference finite-gap solution with $m$ cuts, and let $\{n_1, n_2 \cdots n_m\}$ denote the occupied mode numbers. To excite a mode with an unoccupied $n$, call it $n_*$, the following condition for the quasi-momentum $\mathfrak{p}(\zeta)$ has to satisfy the periodic boundary condition,

$$\mathfrak{p}(\zeta_{m,n_*}) = n_* \pi, \tag{6.5}$$

using $\zeta_{m,n_*}$ to label distinct fluctuation points. Note that these can either be real, or may occur in complex-conjugate pairs (owing to the square-root branch cut nature of the quasi-momentum).

Fluctuation points are treated as "almost degenerate" branch cuts, so small that they do not affect the form of the quasi-momentum $\mathfrak{p}(\zeta)$. This raises an interesting question whether classically any given finite-gap solution is modulationally stable under such fluctuations, see for instance discussions in Ref. [57]. In this respect, we note that the stability condition coincides with the condition for the formation of condensates.

In the following we shall first take a look at the basic case with a single cut. We find the physical contours made out of Bethe roots consist of three pieces: two parts of to the contours which connect to the square-root branch points are joined by a uniform condensate attached to 'the middle', with two additional bent contours emanating from the intersection points that connect to the nearby fluctuation point(s). We give an explicit demonstration of this scenario in Sec. 6.2.1. Presence of multiple excited cuts makes the situation even more involved as cuts exert among themselves an effectively attractive interaction. This situation is described in Sec. 6.2.2.

Let us also mention that a somewhat reminiscent phenomenon is known to appear in the context of matrix models [58] (which are described by a similar type of Riemann-Hilbert problems) and also elsewhere, e.g. in the large-$N$ Yang–Mills theory [59–61] and random tiling models [62, 63]. They commonly go under the name of the Douglas–Kazakov phase transition [61], a variant of a third-order phase transition. Analogous condensates also appear in the semi-classical regime of the Lieb-Liniger model with attractive interaction [64, 65] where a quantum phase transition can be detected through the calculation of correlation functions in

the ground state [65]. We emphasise however that in our case there is no real phase transition going on in the sense that branch points and the quasi-momentum $\mathfrak{p}(\zeta)$ itself do not undergo any discontinuous change, in distinction with the case of Douglas–Kazakov transition where the free energy becomes non-smooth after formation of a "condensate" [63].

### 6.2.1 One-cut case with condensate

In the case of a single cut, we have

$$\int_{\mathcal{C}_1} \mathrm{d}\zeta \rho(\zeta) = \nu_1 \in \mathcal{O}(1), \tag{6.6}$$

with an upper-bounded filling fraction $\nu_1 < 1/2$. Locations of fluctuation points, denote below by $\zeta_{1,k}$, can be inferred from the density

$$\rho_k(\zeta) = \pm \frac{\mathfrak{p}(\zeta) - \pi k}{2\pi i (1 + \delta\zeta^2)}, \qquad \mathfrak{p}(\zeta_{1,k}) = k\pi. \tag{6.7}$$

**Mode number $n = 1$.** The condensate phenomenon can be best illustrated on the basic example of a one-cut solution with mode number $n = 1$. Below the density threshold we find a single smooth arc-shaped contour, as exemplified in panel (a) in Fig. 6. The closest fluctuation point sits at a finite distance away from the cut (somewhere to the left of it), whereas the density at the real axis satisfies

$$\rho(\zeta_*)(1 + \delta\zeta_*^2) < i. \tag{6.8}$$

Increasing the filling fraction will cause an increase in the density on the real axis. During the process, the nearby fluctuation point approaches the physical contour until at the critical filling it eventually collides with it at $\zeta_*$,

$$\rho(\zeta_*)(1 + \delta\zeta_*^2) = i. \tag{6.9}$$

This event is shown in panel (b) in Fig. 6. [8]

Upon increasing the filling fraction even further, the fluctuation points after collision 'tunnel through' the cut. This leaves behind a straight condensate positioned in a vertical direction. The nearest fluctuation point on the real axis will then appear to the right of the physical cut, as pictured in panel (c) in Fig. 6. One can nonetheless recover the same quasi-momentum $\mathfrak{p}(\zeta)$ by considering an additional pair of contours which emanate out of the fluctuation point(s), satisfying $\rho_2(\zeta)\mathrm{d}\zeta \in \mathbb{R}$ with density defined through Eq. (6.7) (depicted by brown dashed lines in panel (c) in Fig. 6). Due to an extra condensate, the original contour cannot accommodate for all the Bethe roots, and some "excess" Bethe roots will lie along those additional contours. We wish to emphasise again that the quasi-momentum $\mathfrak{p}(\zeta)$ remains intact.

There is a suggestive explanation behind the above picture if one pictures a one-cut solution as a limiting (degenerate) case of a more general two-cut solution with one of its cuts 'switched off' to a fluctuation point. This is neatly captured in Fig. 7 in panel (a), where the blue solid lines represent parts of the original physical connecting to the pair branch points $(\zeta_1, \bar\zeta_1)$, while the red dashed line depicts the Bethe-root condensate of uniform density. The extra green solid line belongs to one of the "unphysical contours"[9] associated with the infinitesimal branch cut $(\zeta_F, \bar\zeta_F)$. Combining all the ingredients, we are therefore able to determine the

---

[8]Comparing to the isotropic case in [13], $\nu_c \simeq 0.2092896452$, the condensate appears with a slightly smaller filling fraction.

[9]We call it "unphysical" because the green contour alone does not yield the correct value for the filling fraction for the infinitesimal cut. Yet the combination of all three parts here is clearly physically meaningful.

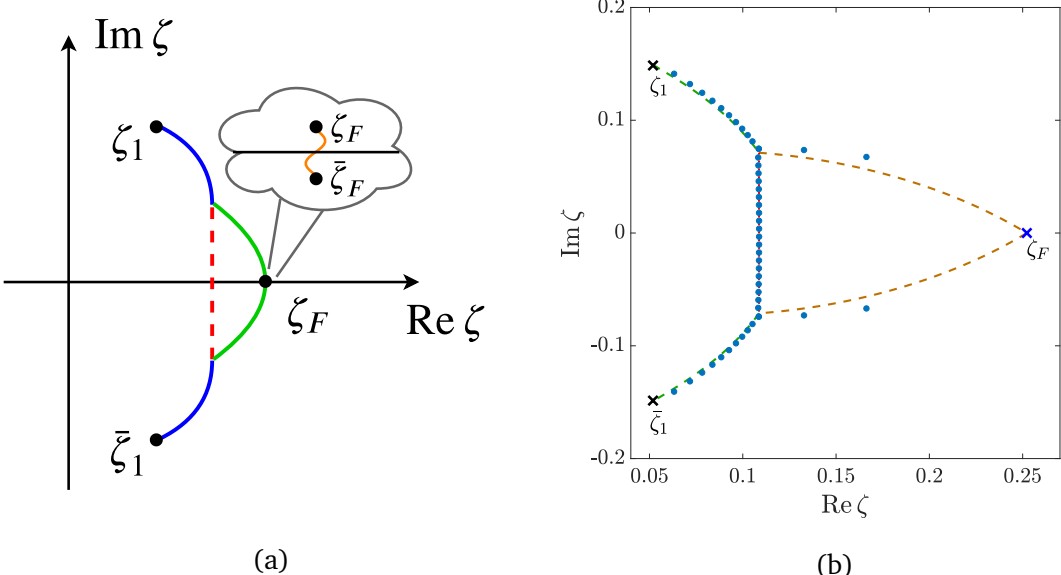

Figure 7: (a) Fluctuation point $\zeta_F$, lying on the real axis, can be viewed as an infinitesimal (collapsed) branch cut – a deactivated mode. (b) Comparison between the classical contour (dashed line) with anisotropic parameter $\delta = 1$, filling fraction $\nu = 0.3$, and mode number $n = 1$ (computed based on reality condition (6.3)) with a condensate (red dashed line) and an additional contour originating from $\zeta_F$ (brown dashed line) and the corresponding solution to the Bethe equations (cf. Eq. (3.3)) with $M = 48$, $L = 144$ and $\eta = \frac{\sqrt{\delta}}{L} = \frac{1}{144}$ (blue dots). Notice that the Bethe roots plotted are $\frac{\tan\vartheta}{\sqrt{\delta}}$.

densities of Bethe roots along these three contours. This amounts to account for the leading-order quantum corrections to ABE (3.16) in non-perturbatively fashion.

To better corroborate the above picture, we made a direct comparison with the contours obtained numerically by solving the Bethe equations for large system sizes, cf. Fig. 7. Moreover, we have performed another quantitative test for the proposed contours through the calculation of the leading-order Gaudin norm for the Bethe states, as shown in Fig. 13. We emphasise that physical contours are a key ingredient for the functional integral approach to compute overlaps (and norms) between semi-classical Bethe eigenstates, thus providing an opportunity to verify whether the described contours are indeed suitable. The numerical evidence is collected in Appendix D.1.

**Mode number $n \geq 2$.** One can encounter even more exotic situations. While $\mathfrak{p}(\zeta_*) = (n+1)\pi$ has only one real solution for $n = 1$, higher mode numbers $n \geq 2$ permit for complex-conjugate pairs of fluctuation points [13]. In this scenario, the same condition $\rho_{n+1}(\zeta)d\zeta \in \mathbb{R}$ yields an additional contour with a condensate appearing between the intersection points, along the lines of the proceeding discussion. This time instead, such contours can be understood at the classical level as arising from a three-cut solution with one large physical cut and two almost degenerate tiny cuts located at the complex-conjugated fluctuation points $\zeta_F$ and $\bar\zeta_F$. For instance, in Fig. 8 we give an illustration of that for the isotropic Heisenberg spin chain with mode number $n = 2$. Unfortunately, for the anisotropic ferromagnet the employed numerical method for producing analogous solutions does not work for $n \geq 2$, cf. Appendix E.1. Given that the distributions of Bethe roots do not appreciably change upon introducing a tiny de-

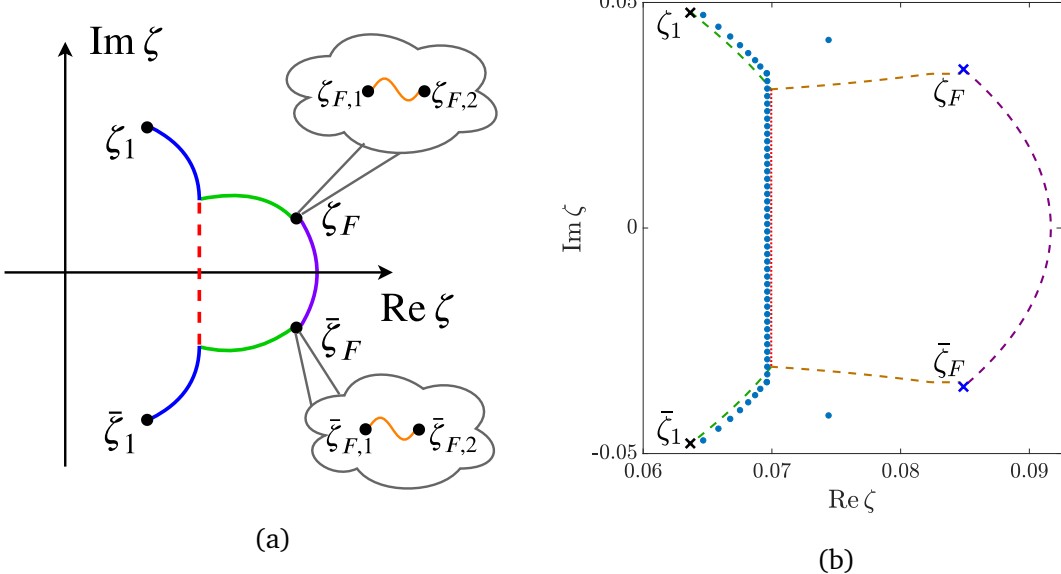

(a)

(b)

Figure 8: (Left) Complex fluctuation points $\zeta_F$ and $\bar{\zeta}_F$. They can be seen as collapsed cuts of a three-cut solution with $\zeta_{F,1} \to \zeta_{F,2} \to \zeta_F$ and $\bar{\zeta}_{F,1} \to \bar{\zeta}_{F,2} \to \bar{\zeta}_F$. (Right) Comparison between the classical density contour (dashed line), including with the condensate (red dashed line) and the additional contours emanating from fluctuation points $\zeta_F$ and $\bar{\zeta}_F$ (brown and purple dashed lines), obtained from reality condition (6.3) for the case of isotropic interaction ($\delta = 0$), with filling fraction $\nu = 0.1$ and mode number $n = 2$, to the corresponding solution to Bethe equations (3.3) with $M = 60$, $L = 600$ and $\eta = 0$. The Bethe roots $\zeta_j$ (blue dots) are rescaled by the system size $L$ and plotted in the inverse spectral plane, i.e. $\zeta_j = 1/(L\lambda_j)$, where $\lambda_j$ solve the isotropic Bethe equations, $\left(\frac{\lambda_j + i/2}{\lambda_j - i/2}\right)^L = \prod_{k \neq j}^M \frac{\lambda_j - \lambda_k + i}{\lambda_j - \lambda_k - i}$.

formation parameter $\eta \sim \mathcal{O}\left(\frac{1}{L}\right)$, we expect the phenomenon to survive the presence of weak interaction anisotropy.

### 6.2.2 Multiple cuts

When multiple cuts get involved, the situation is far more complicated. In that case, the condensates can appear not only out of fluctuation points but also via an attractive interaction amongst the physical cuts. Here we focus for simplicity on the two-cut case, since a general scenario with more cuts can be largely described based on the phenomenology of the two-cut case. In Ref. [13], the authors made an exhaustive survey on the two-cut case at isotropic point ($\delta = 0$). The anisotropic case with $\eta = \frac{\epsilon}{L} > 0$ can be analysed in a similar fashion. There are several discernible features we wish to highlight.

Firstly, when two physical cuts are far apart from one another, each branch cut can produce a condensate upon colliding with their nearby fluctuation points, in analogy with the one-cut case discussed in Ref. [13]. However, when the physical cuts approach closer their mutual attraction amplifies until they eventually merge with one another. The result of this are two joined contours glued via a condensate at the cusps.

A basic instance of the above phenomenon involves two cuts with consecutive mode numbers, namely $n_2 = n_1 + 1$. The moment the two physical contours intersect, say at points $\zeta_{\text{int}}$

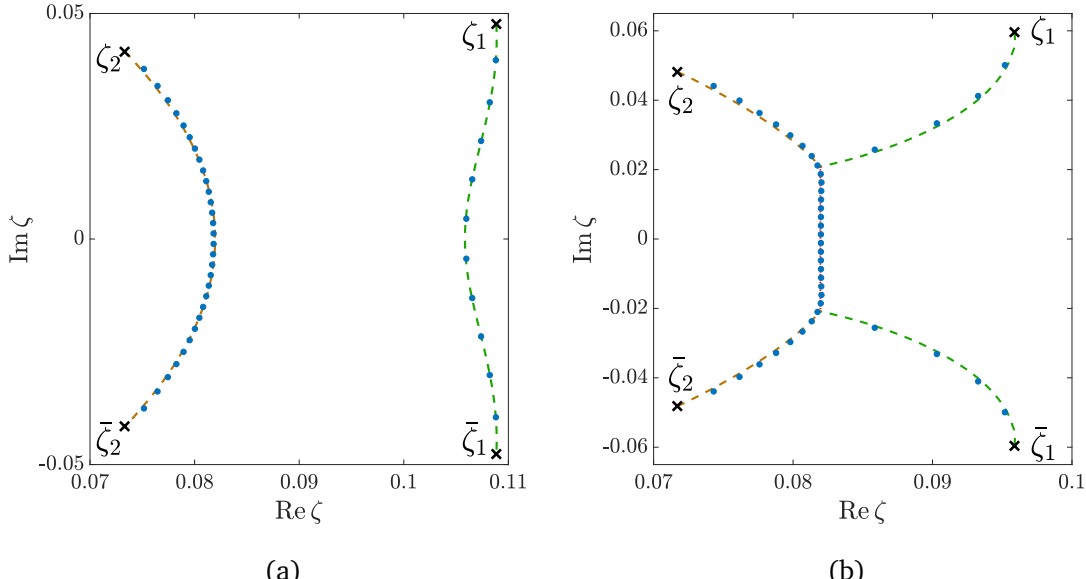

(a)                                         (b)

Figure 9: (Left) Comparison between the density contour (dashed line) of a classical two-cut solution (shown for the isotropic case ($\delta = 0$), with partial filling fractions $\nu_1 = 0.02$, $\nu_2 = 0.06$ and mode numbers $n_1 = 1$, $n_2 = 2$), obtained from reality condition (6.3), and the corresponding numerical solution to Bethe equations (3.3) with $M_1 = 10$, $M_2 = 30$, $L = 500$ and $\eta = 0$ (blue dots). (Right) Comparison between the classical contour (dashed line) (shown for the isotropic case ($\delta = 0$), with partial filling fractions $\nu_1 = 0.025$, $\nu_2 = 0.075$ and mode numbers $n_1 = 1$, $n_2 = 2$), obtained from reality condition (6.3), to the corresponding numerical solution to Bethe equations (3.3) with $M1 = 10$, $M_2 = 30$, $L = 400$ and $\eta = 0$ (blue dots). The Bethe roots plotted are $\zeta_j = 1/(L\lambda_j)$, same as in Fig. 8.

and $\bar{\zeta}_{\text{int}}$, the combined density satisfies

$$\left[\rho_{n_1}(\zeta_{\text{int}}) + \rho_{n_1+1}(\zeta_{\text{int}})\right]\left(1 + \delta\zeta_{\text{int}}^2\right) = \left[\rho_{n_1}(\bar{\zeta}_{\text{int}}) + \rho_{n_1+1}(\bar{\zeta}_{\text{int}})\right]\left(1 + \delta\bar{\zeta}_{\text{int}}^2\right) = i, \qquad (6.10)$$

giving birth to a condensate. Indeed, installing a condensate between the two such intersection points does not alter the the quasi-momentum and hence the filling fraction stays intact. We have confirmed this to be the case by numerically solving the Bethe equations for moderately large system sizes, as demonstrated in Fig. 9 (again for the isotropic case). In particular, at low filling fractions for both cuts their mutual "attraction" becomes apparent (cf. the second cut connecting $(\zeta_1, \bar{\zeta}_1)$ in panel (a) in Fig. 9). The four branch points in panel (a) in Fig. 9, reading $\zeta_1 = 0.10884679 + 0.047665716i$ and $\zeta_2 = 0.07330641 + 0.04152184i$ (with filling fractions $\nu_1 = 0.02$, $\nu_2 = 0.06$, $\ell = 1$ and mode numbers $n_1 = 1$, $n_2 = 2$), have been determined numerically using the recipe given in Appendix E.2. At a certain critical value of the filling fractions the two cuts merge together. The intersection point becomes a logarithmic branch point of a condensate, as exemplified in panel (b) in Fig. 9. The four branch points in panel (b) in Fig. 9 are $\zeta_1 = 0.09587725 + 0.05961115i$ and $\zeta_2 = 0.07169871 + 0.04814544i$ with filling fractions $\nu_1 = 0.025$, $\nu_2 = 0.075$, $\ell = 1$ and mode numbers $n_1 = 1$, $n_2 = 2$. For the anisotropic interaction we encountered the same numerical difficulties as previously for the one-cut solution with $n \geq 2$, cf. Appendix E.1. We nevertheless do not expect any qualitative difference compared to the isotropic model.

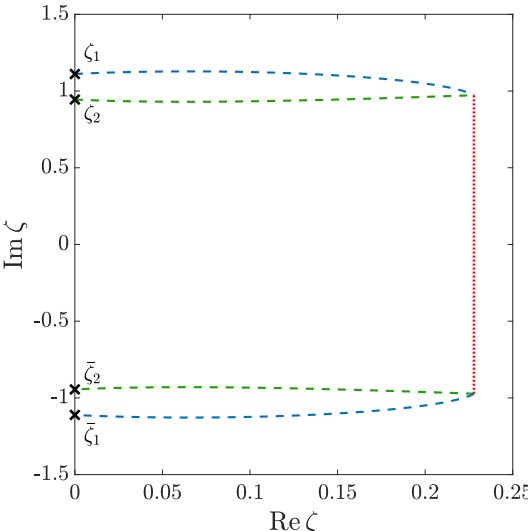

Figure 10: Quantised bion configuration with the proposed density contours (physical cuts). The two nonlinear modes with $\delta = 1$, parametrised by pairs of branch points $(\zeta_1, \bar{\zeta}_1)$ and $(\zeta_2, \bar{\zeta}_2)$ on imaginary $\zeta$-axis, have associated mode numbers $n_1 = 1$ and $n_2 = -1$. The branch points have been found numerically and are located at $\zeta_1 = (1/0.9)$i and $\zeta_2 = (1/1.058355921)$i, The red dashed line represents the "double condensate". The corresponding classical solution is plotted in Fig. 4.

## 6.3 Special case: bion

As discussed earlier, the easy-axis regime (i.e. for $\delta > 0$) permits for a distinguished subclass of two-cut solutions that do not take place in the other two (that is isotropic and easy-plane) regimes. Here we have in mind the bion solution which we have described and parametrised in Sec. 5.2. One part of the motivation for investigating this case in detail is to elucidate the microscopic origin and stability of kinks in Landau–Lifshitz field theory, which we expect to have a pivotal importance for understanding the freezing property of a domain-wall profile, investigated recently in Refs. [16, 17].

To quantise the classical bion solution we demand the same reality condition as previously in the one-cut case. Notice however that bion solutions belong to maximally saturated states with the total filling $\nu = \frac{1}{2}$. Condensates appear to be a common feature at half filling. In describing a bion solution, there is no loss of generality in fixing the mode numbers of the two cuts to $\pm 1$, such that $\Delta n = 2$. Recall that in a general situation with two cuts being far apart, each cut can grow a condensate on its own. The bion case is different in that the two branch cuts reside close to each other and share a condensate in common. In fact, in the isotropic ferromagnet studied in Refs. [13, 56] the two-cut solution with mode numbers set to $\pm 1$ is known to result in a "double condensate". Led by this observation, we conjecture that the same phenomenon takes place presently in the case of bions; a "double condensate" emerges when a pair of branch cuts with mode numbers $\pm 1$ intersect with one another, thereby producing a logarithmic cut with of 'doubled' uniform density 2i. Analogous objects which are twice as dense as ordinary Bethe strings have been previously found in Ref. [56] in the study of solutions to the isotropic Bethe equations.

We proceed by semi-classically quantising the bion solution using the conjectured form of its contours with a double condensate, depicted in Fig. 10 by the red dashed line satisfying

$$\rho(\zeta)(1 + \delta\zeta^2) = 2\mathrm{i}. \tag{6.11}$$

We have been able to explicitly match the classical values of the filling fraction, momentum and energy: the two partial filling fractions add up exactly to one half, i.e. $\nu = \nu_1 + \nu_2 = \frac{1}{2}$, whereas total momentum $P = 0 \,(\mathrm{mod}\, 2\pi)$ and total energy $E = 3.96045$ (obtained by numerically integrating along the proposed contours) match those of a classical bion configuration, with $P = 0$ and $E = 3.960358(6)$. These results strongly indicate that we have indeed correctly identified the physical contours associated to a quantised bion.

As discussed earlier in Section 5.2, kinks arise as a particular (soliton) limit of a bion solution in which the two branch points $\zeta_1, \zeta_2$ on the imaginary axis coalesce at $\mathrm{i}/\epsilon$. By inspecting this degeneration process at the level of semi-classical eigenstates, we find a uniform condensate with a double density of Bethe roots running along the imaginary axis between $-\mathrm{i}/\epsilon$ and $\mathrm{i}/\epsilon$. We note that (anti)kinks are not compatible with periodic boundary conditions. In an infinitely extended quantum chain however, the kink and antikink eigenstates represent extra degenerate ground states (with broken translational symmetry) of the XXZ ferromagnet in the gapped phase. Kink eigenstates have been derived in Refs. [66, 67] using a curious correspondence between the XXZ quantum chain and the problem of a melting crystal corner. This derivation however does not require any use of the Bethe quantisation condition and consequently cannot reveal the internal magnon structure of the kink. It would be valuable to devise a method to extract the corresponding numerical solution to the Bethe equations for large system sizes. The task of solving the anisotropic Bethe equations (3.3) in the vicinity of half filling remains quite challenging at this moment. Perhaps one could get some hints by first scanning through the complete list of exact eigenstates for relatively small system sizes (typically of order $L \sim 10$, using e.g. the techniques proposed in Refs. [68, 69]) and look for traces of finite-size bions. At this junction, our statements regarding the kink solution thus remain to an extent conjectural.

# 7 Semi-classical norms and overlaps

In this section we outline how to compute an overlap between two semi-classical Bethe eigenstates. We provide closed formulae for (i) the Gaudin norm and (ii) the Slavnov inner product between an on-shell and off-shell Bethe states [70]. There are two possible routes to achieve this. The first one, proposed by Gromov et al. in [31], is to perform coarse-graining directly at the level of the general determinant formula for a specific finite-gap density resolvent. The other approach, developed for the isotropic (XXX) Heisenberg model by Kostov and Bettelheim in Refs. [32–34], makes use of functional integration techniques with a bit of complex analysis. Both methods provide the leading (i.e. classical) contributions to the overlaps and norms. We shall not repeat the derivations here but instead only succinctly summarise the main formulae for the model of our interest. Moreover, in Sections D.1 and D.2 we provide a direct numerical confirmation based on the finite-size analysis.

## 7.1 Gaudin norm

The method proposed by Kostov in Refs. [32, 33] has already been generalised for the specific case of the anisotropic Heisenberg model in [71, 72]. To compute the Gaudin norm we instead employ a more direct approach of [31], which we generalise here by including the interaction anisotropy. The idea is to convert the logarithm of the Gaudin determinant into a Riemann summation which, after taking the $L \to \infty$ limit, corresponds to complex integration along the physical contours which support the Bethe roots.

The Gaudin norm of a finite-volume Bethe eigenstate,

$$\mathcal{N} = \langle \{\vartheta\} | \{\vartheta\} \rangle, \tag{7.1}$$

grows exponentially in system size, i.e. $\log \mathcal{N} \sim \mathcal{O}(L)$ to the leading order of $L$. In the $\zeta$-plane parametrisation, we find the following explicit form

$$
\log \mathcal{N} = C_1 L + o(L),
$$
$$
C_1 = \int_{\mathcal{C}} d\zeta \left[ i\pi\ell(1 + \delta \zeta^2)\rho(\zeta) + 2 \int_0^{\rho(\zeta)(1+\delta \zeta^2)} \frac{d\xi}{1 + \delta \xi^2} \log\big((2\sinh(\pi\xi))\big) \right], \tag{7.2}
$$

where we have expressed the volume-law coefficient $C_1$ in terms of the resolvent density $\rho(\zeta)$ with support on a union of contours $\mathcal{C} = \cup_j \mathcal{C}_j$.

We note that the dominant subleading correction to the above formula is quite subtle and has the form $\log \mathcal{N}(L) = C_1 L + \frac{1}{2}\log L + \mathcal{O}(L^0)$ (for $C_1 \in \mathcal{O}(1)$), as discussed in Ref. [31]. Several numerical verifications (both with or without a condensate) are presented in Appendix D.1.

## 7.2 Slavnov overlap

To express the semi-classical overlaps we follow instead the functional integral approach devised in Refs. [32, 33]. This method does not rely on the clustering properties of Gaudin determinant as in Ref. [31]. Here we merely quote the final result of [71] (cf. formula (1.5) in there) for the anisotropic Landau–Lifshitz field theory

$$
\log\langle\{\vartheta_1\}|\{\vartheta_2\}\rangle \simeq \oint_{\mathcal{C}_{1\cup2}} \frac{d\zeta}{2\pi i} \log \Phi_{\sqrt{\eta}}\big(\mathfrak{p}_1(\zeta) + \mathfrak{p}_2(\zeta) + \pi\big), \tag{7.3}
$$

involving two classical quasimomenta $\mathfrak{p}_1(\zeta)$ and $\mathfrak{p}_2(\zeta)$ that correspond to the semi-classical Bethe eigenstates $|\{\vartheta_1\}\rangle$ and $|\{\vartheta_2\}\rangle$ [10], respectively. Function $\Phi_b(z)$ stands for quantum dilogarithm [73], defined through the following integral representation

$$
\Phi_b(z) = \exp\left[ \frac{i}{2} \int_{\mathbb{R}+i0} \frac{dt}{t} \frac{e^{zt}}{\sin(b^2 t)\sinh(\pi t)} \right]. \tag{7.4}
$$

This function can be understood of as a 'quantum deformation' [11] of the ordinary dilogarithm function $\mathrm{Li}_2(z)$ to which it reduces in the isotropic limit $\delta \to 0$. The contour prescription in Eq. (7.3) is such that $\mathcal{C}_{1,2}$ wrap around tightly around the supports of the respective density resolvents, cf. Ref. [33].

Further simplification of the above formula can be made in the semi-classical limit $\eta \to \epsilon/L$ which implies $b = \sqrt{\eta} \to 0$. In this limit the quantum dilogarithm simplifies to [74] [12]

$$
\lim_{b\to 0} \Phi_b(z) = \exp\left[ \frac{iL}{\sqrt{\delta}} \mathrm{Li}_2(-e^{iz}) \right] + \mathcal{O}(L^0), \tag{7.5}
$$

and accordingly at the leading order $\mathcal{O}(L)$ the logarithmic overlap is approximately

$$
\lim_{\eta\to\epsilon/L} \log\langle\{\vartheta\}_1|\{\vartheta\}_2\rangle = C_2 L + o(L),
$$
$$
C_2 = \frac{1}{\sqrt{\delta}} \oint_{\mathcal{C}_{1\cup2}} \frac{d\zeta}{2\pi(1 + \delta \zeta^2)} \mathrm{Li}_2\left[ e^{i\big(\mathfrak{p}_1(\zeta)+\mathfrak{p}_2(\zeta)\big)} \right]. \tag{7.6}
$$

---

[10] Only one of the states has to be on-shell, i.e. solution to Bethe ansatz equations (3.3).

[11] The word 'quantum' here refers to the $q$-deformation parameter of 'quantum calculus' which should not be confused with the $q$-parameter of the quantum algebra $\mathcal{U}_q(\mathfrak{su}(2))$ of the underlying anisotropic Heisenberg chain.

[12] Beware that the definition of $\Phi_b(z)$ in [74] differs from the definition in [71] and the one used here.

Similarly to the Gaudin norm, the dominant subleading correction is of form $\log\langle\{\vartheta\}_1|\{\vartheta\}_2\rangle = C_2 L + \frac{1}{2}\log L + \mathcal{O}(L^0)$.

For the coinciding sets of rapidities, this correctly reproduces the leading order expression for the Gaudin norm,

$$\lim_{\eta\to\epsilon/L}\log\langle\{\vartheta\}|\{\vartheta\}\rangle \simeq \frac{L}{\sqrt{\delta}}\oint_{\mathcal{C}}\frac{\mathrm{d}\zeta}{\pi(1+\delta\,\zeta^2)}\mathrm{Li}_2\left(e^{2i\mathfrak{p}(\zeta)}\right), \tag{7.7}$$

which can be readily reconciled with Eq. (7.2) upon expressing the resolvent density in terms of the quasi-momentum, $i\pi\rho(\zeta) = \pi n - \mathfrak{p}(\zeta)$, and performing the following integral

$$\int_0^{\rho(\zeta)(1+\delta\zeta^2)}\frac{\mathrm{d}\xi}{1+\delta\xi^2}\log\left(2\sinh(\pi\xi)\right) = \frac{1}{2\pi}\mathrm{Li}_2\left(e^{2i\mathfrak{p}(\zeta)}\right) + \frac{\pi}{2}\rho^2(\zeta)(1+\delta\zeta^2)^2 - \frac{\pi}{12}. \tag{7.8}$$

There is a practical limitation of Eq. (7.6) that concerns the placement of integration contours, acknowledged previously in Ref. [33]. The requirement is that the integration contours $\mathcal{C}$ must avoid crossing any branch cut of the function in the integrand. This issue is presently further complicated by additional logarithmic branch cuts due to the dilogarithm function. This shortcoming makes the numerical verification a challenging task. We nonetheless still managed to verify its validity in special case of overlaps with a vacuum descendant (see Appendix D.2).

## 8 Correlation functions

We have thus far demonstrated that the knowledge of physical contours not only proves useful in calculating the spectrum, overlaps and norms of semi-classical Bethe eigenstates, but also facilitates the semi-classical quantisation of the finite-gaps solutions. On the other hand, we have not yet addressed the expectation values of physical observables. This section is devoted to discuss some properties of correlation functions at the semi-classical level.

Despite integrability, the task of computing exact expectation values of physical observables, including their correlation functions, appears quite challenging. There have already been numerous works on this subject, employing either the form-factor expansion or bootstrap methods, e.g. [38, 75–78]. Here, however, we are particularly interested in the semi-classical regime where those methods are not directly applicable.

We are specifically interested whether the aforementioned classical–quantum correspondence for correlations functions, established analytically in the introductory section 2 on the toy example of the harmonic oscillator, holds on more general grounds. A direct generalisation of this principle from a single-particle paradigm is complicated by the fact that integrable field theories governed by PDEs involve many degrees of freedom which, moreover, couple (i.e. interact) in a non-trivial fashion. One viable empirical approach to obtain correlation functions in classical regime (enabling a comparison with their quantum counterparts) is to build on the semi-classical form-factor approach developed by Smirnov [79]. Accordingly, the matrix elements would be represented as integrals over $\gamma$ variables, see Eqs. (33)–(36) in [79].

Let us consider a periodic solution associated with one branch cut. It is described by a single dynamic variable $\gamma$. We further replace the integration over variable $\gamma$ with the integration over one period, similar to the phase space averaging [80]. In addition, we compute numerically (for small system sizes) the quantum correlators in the eigenstate that corresponds to the cut.

We focus only on static (i.e. equal-time) correlation functions, considering one-point functions $\langle\hat{\sigma}_j^x\rangle$ and $\langle\hat{\sigma}_j^z\rangle$ and two-point point functions $\langle\hat{\sigma}_j^x\hat{\sigma}_k^x\rangle$ and $\langle\hat{\sigma}_j^z\hat{\sigma}_k^z\rangle$. Specifically, we set the

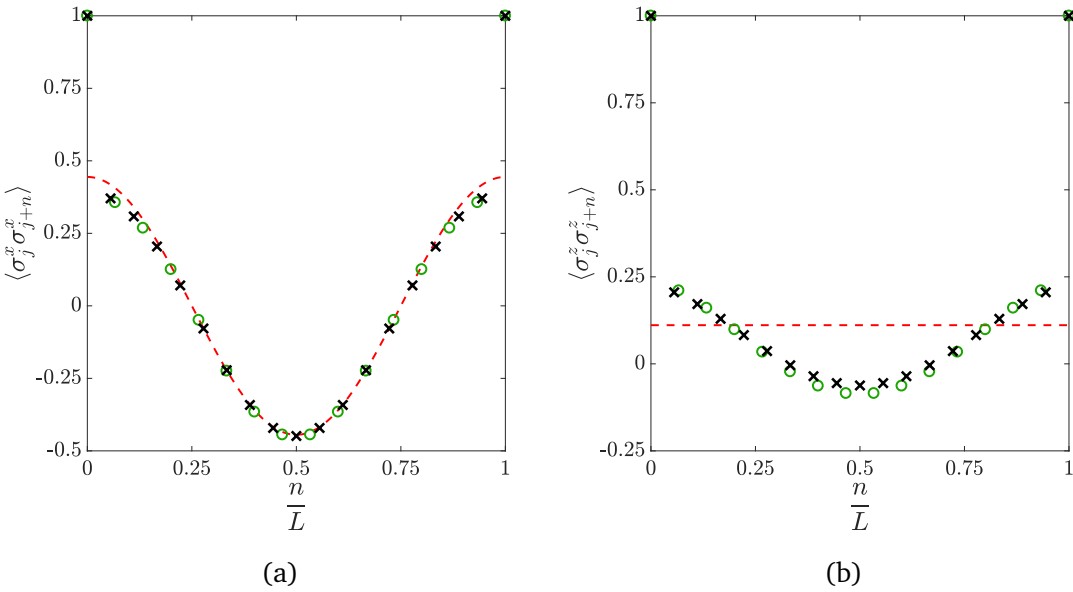

Figure 11: (a) Quantum correlation functions $\langle \hat{\sigma}_j^{\mathrm{x}} \hat{\sigma}_{j+n}^{\mathrm{x}} \rangle$ (shown in green circles for $L = 15$ and black crosses for $L = 18$) versus averaged classical correlation functions (red dashed line), with anisotropy $\delta = 1$ and classical period $\ell = 1$. (b) The same comparison for longitudinal correlators $\langle \hat{\sigma}_j^{\mathrm{z}} \hat{\sigma}_{j+n}^{\mathrm{z}} \rangle$.

filling fraction to $\nu = \frac{1}{3}$ and the mode number to $n = 1$. We present our results for system sizes $L = 15$ and $18$ with the with number of down-turned spins being 5 and 6, respectively. The corresponding averages are denoted as $\langle \cdots \rangle_5$ and $\langle \cdots \rangle_6$. The coordinate Bethe ansatz wavefunctions can be represented in the local eigenstate basis of $\hat{\sigma}^{\mathrm{z}}$ by solving Bethe ansatz equations (3.3). An important thing to keep in mind is that quantum states (wavefunctions) are eigenfunctions of the momentum operator and consequently expectations values of one-point observables do not have any dependence on the spatial coordinate (lattice index). Two-point functions on the other hand can only be a function of the distance. In contrast, classical spin field configurations exhibit non-uniform dependence on the spatial coordinate $x$ and can be thus compared to the quantum correlation functions evaluated on semi-classical eigenstates after an appropriate phase-space averaging. In particular, for a periodic classical spin-field configuration with period $\ell$ and an operator (product of operators) $O[\mathcal{S}]$ that functionally depends on the spin configuration $\mathcal{S}$, we expect the corresponding correlation functions to take the form

$$\langle \{\vartheta\} | O[\mathcal{S}] | \{\vartheta\} \rangle \simeq \langle O[\mathcal{S}] \rangle_{\mathrm{cl}} = \frac{1}{\ell} \int_0^\ell \mathrm{d}x \, O[\mathcal{S}(x)], \tag{8.1}$$

with $|\{\vartheta\}\rangle$ denoting the corresponding semi-classical quantum eigenstate in the thermodynamic limit $L \to \infty$. The classical spin configuration for $n = 1$ and $\nu = \frac{1}{3}$ reads

$$\mathcal{S}^{\mathrm{z}}(x, t) = 1 - 2\nu = \frac{1}{3},$$

$$\mathcal{S}^{\mathrm{x}} = 4(\nu - \nu^2)\cos(kx + wt) = \frac{8}{9}\cos\left[\frac{2\pi}{\ell}x + \frac{1}{3}\left(\frac{4\pi^2}{\ell^2} + \delta\right)t\right]. \tag{8.2}$$

Therefore for the one-point functions, we find

$$\langle \hat{\sigma}_j^{\mathrm{x}} \rangle_5 = \langle \hat{\sigma}_k^{\mathrm{x}} \rangle_6 = 0 = \frac{1}{\ell} \int_0^\ell \mathrm{d}x \, \mathcal{S}^{\mathrm{x}}(x), \qquad \langle \hat{\sigma}_j^{\mathrm{z}} \rangle_5 = \langle \hat{\sigma}_k^{\mathrm{z}} \rangle_6 = \frac{1}{\ell} \int_0^\ell \mathrm{d}x \, \mathcal{S}^{\mathrm{z}}(x) = \frac{1}{3}, \tag{8.3}$$

for $j = 1, 2, \ldots 15$ and $k = 1, 2, \ldots, 18$, thus confirming the correspondence.

Our results for two point functions are presented in Fig. 11. In this case we notice some discrepancies between the classical expectation values,

$$\langle \hat{\sigma}_j^{\text{x}} \hat{\sigma}_{j+n}^{\text{x}} \rangle \simeq \frac{1}{\ell} \int_0^\ell \mathrm{d}x \, \mathcal{S}^{\text{x}}(x) \mathcal{S}^{\text{x}}\left( x + \frac{n\ell}{L} \right) = 2(\nu - \nu^2) \cos\left( \frac{2\pi n}{L} \right), \tag{8.4}$$

$$\langle \hat{\sigma}_j^{\text{z}} \hat{\sigma}_{j+n}^{\text{z}} \rangle \simeq \frac{1}{\ell} \int_0^\ell \mathrm{d}x \, \mathcal{S}^{\text{z}}(x) \mathcal{S}^{\text{z}}\left( x + \frac{n\ell}{L} \right) = (1 - 2\nu)^2, \tag{8.5}$$

and their quantum counterparts, which we attribute to the finite number of spins. Indeed, instead of having a condensed contour of Bethe roots representing the branch cut in the complex plane, we consider solutions with at most 6 Bethe roots. We nonetheless find it plausible that with increasing system sizes the deviations would gradually diminish and we therefore expect to recover the classical result in the $L \to \infty$ limit. Finite-size effects also depend on type of operators that appear in the correlator; in the case of $\langle \hat{\sigma}_j^{\text{z}} \hat{\sigma}_{j+n}^{\text{z}} \rangle$ the deviation from the asymptotic result is found to be larger than in the case of $\langle \hat{\sigma}_j^{\text{x}} \hat{\sigma}_{j+n}^{\text{x}} \rangle$, see Fig. 11. Here we have demonstrated the first step in understanding such correlation functions. A systematic and comprehensive numerical analysis of the correspondence and finite-size corrections is postponed for future work.

# 9 Conclusion and outlook

We have studied the structure of the semi-classical spectrum of the anisotropic Heisenberg spin-1/2 chain in the easy-axis regime with weak anisotropies. Using the framework of algebraic integrability, we have established that these semi-classical eigenstates emerge classically as interacting nonlinear spin waves governed by the Landau–Lifshitz field theory with uniaxial anisotropy. Firstly, we have expressed the asymptotic Bethe equations in the form of a singular integral equation for the spectral resolvent, which we subsequently recast in the form of the Riemann–Hilbert problem, providing jump discontinuity conditions for a double-valued complex function called quasi-momentum. The latter encodes the moduli of hyperelliptic Riemann surfaces which provide complete information about the spectrum of nonlinear eigenmodes for a class of finite-gap solutions of the anisotropic Landau–Lifshitz ferromagnet.

We have outlined the main ingredients of the algebro-geometric integration technique. The starting point of this approach is the usual Lax representation which realises an auxiliary linear problem of parallel transport on a smooth manifold with a flat connection. In our formulation we made use of the adjoint representation, enabling us to parametrise the solutions in terms of squared polynomial eigenfunctions whose zeros contain information about the dynamical phases evolving on a finite-genus Riemann surface. We have demonstrated how their dynamics can be mapped to a linear evolution on the Liouville hypertorus using the Abel–Jacobi transform. Finally, individual components of the physical spin field can be retrieved with aid of reconstruction formulae.

We have implemented the finite-gap integration procedure for two simplest classes of solutions: (i) the single-mode (one-cut) solutions, describing precessional motion around the anisotropy axis, and (ii) the two-mode (two-cut) solutions which take the shape of elliptic waves. Amongst the two-cut solutions, there are special elliptic solutions that describe bions, a bound state of kink and antikink. In a particular singular limit, the bion solution degenerates into the static kink.

One central result of our work is an algorithm for performing semi-classical quantisation of classical finite-gap solutions. In this respect, the key object is the density resolvent asso-

ciated to the classical quasi-momentum. The spectral resolvent is supported on a union of one-dimensional segments in the complex plane which may be adopted as branch cuts of a Riemann surface of finite genus. By following the programme of Ref. [13], we described and implemented a numerical algorithm for determining the locations of physical cuts (associated with the density contours). Each branch cut is a magnon condensate that represents a non-linear mode in the spectrum of the effective classical equation of motion. In this view, semi-classical quantisation amounts to dissolve each branch cut of a finite-gap quasi-momentum into a large but finite number of magnetisation quanta (carried by magnons) with a prescribed accuracy; the resolvent density along each contour specifies a local density of magnon excitations. When the local density of magnons exceeds a critical threshold value, the physical contours experience a certain 'non-perturbative effect' which leads to the formation of special condensates with uniform unit density of Bethe roots. A proper resolution of such situations necessitates to take into account quantum corrections.

In this work we devote most attention to various formal properties of semi-classical eigenstates and other related mathematical underpinnings. We hope this can provide a foundation for further developments which would ultimately pave the way to physical applications. Particularly in the domain of out-of-equilibrium dynamics there has been tremendous progress recently in employing integrability techniques that enabled us, among others, to study late-time relaxation dynamics from highly-excited many-body initial states which goes commonly under the name of 'quantum quenches' [3], see also [81–83]). A particularly useful tool in this regard is the functional integral representation, dubbed the Quench Action [84, 85], which exploits exact knowledge of thermodynamic overlap coefficients. Our hope is to obtain its semi-classical counterpart. Despite that general expressions for the overlap coefficients between a semi-classical eigenstates and an off-shell state are explicitly known due to Ref. [32, 33, 71], we have not been successful in employing them in practice yet. We have nonetheless been able to provide several benchmarks for the computation of Gaudin norms and Slavnov overlaps for a few simple finite-gap solutions and found good convergence.

The main difficulty when dealing with the overlap formulae was to satisfy the requirements for the contour prescription. Since avoiding all the branch cuts of the integrand does not appear to be easily overcome, it seems that an alternative formula based solely on the resolvent densities (similarly to that for the semi-classical limit of the Gaudin norm [31]) might be preferable. Overcoming this issue would be a stepping stone for formulating a quench problem at the level of semi-classical states, a prominent example of which would be the semi-classical version of the domain wall melting which has recently been solved analytically by the authors in [16] using the inverse scattering transformation. This could help to solidify the classical–quantum correspondence also brought forward in [16] and corroborated in [17].

Lastly, we shortly examined the structure of correlation functions in the semi-classical eigenstates of the Heisenberg XXZ chain and compared them to their classical counterparts, namely correlators of classical fields as finite-gap solutions. We found empirical evidence for a classical–quantum correspondence between static multipoint correlators on both sides, in alignment with the earlier results of Ref. [79]. Importantly, the semi-classical correlators can only be compared to correlators of classical fields after computing phase-space averages, as demonstrated on a few basic examples. While there are strong indications that such a correspondence should hold generally in quantum integrable models that possess (integrable) classical limits, a proof is still lacking at the moment. We believe that it would be fruitful to investigate this matter in the framework of quantum separated variables, see e.g. [53, 86, 87], to learn how classical separated variables on Riemann surfaces [36] emerge from the microscopic quantum model that sits underneath. In our opinion, quantum integrable lattice models and spin chains provide paradigmatic examples to address these aspects.

## Acknowledgements

We are very grateful to Jean-Sébastien Caux for his involvement at the starting stage of the project and numerous discussions. We thank Filippo Colomo, Andrea De Luca, Andrii Liashyk, Grégoire Misguich, Vincent Pasquier, and Dmytro Volin for valuable discussions. We are greatly indebted to Nikolay Gromov and Ivan Kostov for sharing their insights on various facets of the problem. Y.M. thanks Vincenzo Alba and Alvise Bastianello for useful suggestions on numerical implementations.

Y.M., and O.G. acknowledge the support from the European Research Council under ERC Advanced grant 743032 DYNAMINT. E.I. is supported by the research programme P1-0402 of Slovenian Research Agency.

## A   Riemann-Hilbert problem in $\zeta$-plane

In order to study the formation of condensates, the Riemann–Hilbert problem is most conveniently written in terms of spectral parameter $\zeta = 1/\mu$, namely

$$\mathfrak{p}(\zeta + i0) + \mathfrak{p}(\zeta - i0) = 2\pi n_j, \quad \zeta \in \mathcal{C}_j, \tag{A.1}$$

where $\mathcal{C}_j$ denotes the $j$-th branch cut in $\zeta$ plane, whereas quasi-momentum $\mathfrak{p}(\zeta)$ is defined as

$$\mathfrak{p}(\zeta) = G(\zeta) - \frac{\ell}{2\zeta} = \ell \int d\xi \tilde{\mathcal{K}}_\delta(\zeta, \xi)\rho(\xi) - \frac{\ell}{2\zeta}, \tag{A.2}$$

with integration kernel

$$\tilde{\mathcal{K}}_\delta(\zeta, \xi) = \frac{1 + \delta \xi \zeta}{\zeta - \xi}. \tag{A.3}$$

The density (of the Bethe roots) is accordingly given by

$$\rho(\zeta) = \frac{\mathfrak{p}(\zeta + i0) - \mathfrak{p}(\zeta - i0)}{2i\pi\ell(1 + \delta\zeta^2)}, \quad \zeta \in \mathcal{C}_j. \tag{A.4}$$

Note that the orientation of integration along $\mathcal{C}_j$ is now in the opposite direction as previously, i.e. it goes from the branch point with negative imaginary part to the one with positive imaginary part.

## B   Finite size corrections to Riemann-Hilbert problem

Here we outline how to take the semi-classical limit of the logarithm of $Q_j^{[\pm 2]}$. The first step is to split the term into the anomalous part and normal part [55,88], i.e.

$$
\begin{aligned}
\log Q_j^{[\pm 2]}(\vartheta_j) &= \sum_{k \neq j}^{M} \log \sin(\vartheta_j - \vartheta_k \pm i\eta) \\
&= \sum_{0 < |k-j| \leq K} \log \sin(\vartheta_j - \vartheta_k \pm i\eta) + \sum_{|k-j| > K} \log \sin(\vartheta_j - \vartheta_k \pm i\eta),
\end{aligned}
\tag{B.1}
$$

where parameter $K$ is a cut-off with the following properties,

$$\vartheta_j - \vartheta_k \sim \begin{cases} \mathcal{O}(1/L), & |k-j| \leq K \\ \mathcal{O}(1), & |k-j| > K \end{cases}. \tag{B.2}$$

We denote the anomalous part as

$$\log Q_j^a(\vartheta_j \pm i\eta) = \sum_{0 < |k-j| \le K} \log \sin(\vartheta_j - \vartheta_k \pm i\eta), \tag{B.3}$$

while the normal part is

$$\log Q_j^n(\vartheta_j \pm i\eta) = \sum_{|k-j| > K} \log \sin(\vartheta_j - \vartheta_k \pm i\eta). \tag{B.4}$$

For the normal part, we can perform the same expansion as in Eq. (3.12), namely

$$\log Q_j^n(\vartheta_j \pm i\eta) = \log Q_j^n(\vartheta_j) \pm i\eta \frac{d}{d\vartheta} \log Q_j^n(\vartheta)|_{\vartheta=\vartheta_j}$$
$$- \frac{\eta^2}{2} \frac{d^2}{d\vartheta^2} \log Q_j^n(\vartheta)|_{\vartheta=\vartheta_j} + \mathcal{O}\left(\frac{1}{L^2}\right), \tag{B.5}$$

and

$$i\eta \frac{d}{d\vartheta} \log Q_j^n(\vartheta)|_{\vartheta=\vartheta_j} = \frac{\epsilon\ell}{L} \sum_{|k-j|>K} \frac{1}{\tan(\vartheta_j - \vartheta_k)} = \frac{\ell}{L} \sum_{|k-j|>K} \frac{\mu_j \mu_k + \delta}{\mu_j - \mu_k}. \tag{B.6}$$

Combining the two parts, we obtain

$$\log Q_j^n(\vartheta_j + i\eta) - \log Q_j^n(\vartheta_j - i\eta) = \frac{2\ell}{L} \sum_{|k-j|>K} \frac{\mu_j \mu_k + \delta}{\mu_j - \mu_k} + \mathcal{O}\left(\frac{1}{L^2}\right). \tag{B.7}$$

Meanwhile, for the anomalous part, denoting $m = k - j$, we have

$$\log Q_j^n(\vartheta_j + i\eta) - \log Q_j^n(\vartheta_j - i\eta) = \sum_{0 < |m| < K} \log \frac{\sin(\vartheta_j - \vartheta_{j+m} + i\eta)}{\sin(\vartheta_j - \vartheta_{j+m} + i\eta)}$$
$$= \sum_{0 < |m| < K} \log \frac{L(\mu_j - \mu_{j+m}) + i\ell(\mu_j^2 + \delta)}{L(\mu_j - \mu_{j+m}) - i\ell(\mu_j^2 + \delta)}. \tag{B.8}$$

We can develop an expansion

$$L\mu_{j+m} \sim c_1 L + c_2 m + \frac{1}{2}\frac{c_3 m^2}{L} + \mathcal{O}\left(\frac{1}{L^2}\right), \quad |m| \le K, \tag{B.9}$$

where all the "constants" can be expressed in terms of density $\rho(\mu)$, i.e.

$$c_1 = \mu_j, \quad c_2 = \frac{1}{\rho(\mu_j)}, \quad c_3 = -\frac{\rho'(\mu_j)}{\rho(\mu_j)^3}, \tag{B.10}$$

and

$$\rho(\mu) = \frac{1}{L} \sum_{j=1}^{M} \delta(\mu - \mu_j), \quad \rho(\mu) \simeq \frac{dj}{d\mu}. \tag{B.11}$$

By combining the $m$-th and $(-m)$-th terms in the sum, we can express the leading order of the sum as

$$\sum_{m=1}^{K} \frac{1}{i}\left(\log \frac{L(\mu_j - \mu_{j-m}) + i\ell(\mu_j^2 + \delta)}{L(\mu_j - \mu_{j-m}) - i\ell(\mu_j^2 + \delta)} + \log \frac{L(\mu_j - \mu_{j+m}) + i\ell(\mu_j^2 + \delta)}{L(\mu_j - \mu_{j+m}) - i\ell(\mu_j^2 + \delta)}\right), \tag{B.12}$$

using

$$\frac{1}{i}\left(\log\frac{L(\mu_j-\mu_{j-m})+i\ell(\mu_j^2+\delta)}{L(\mu_j-\mu_{j-m})-i\ell(\mu_j^2+\delta)}+\log\frac{L(\mu_j-\mu_{j+m})+i\ell(\mu_j^2+\delta)}{L(\mu_j-\mu_{j+m})-i\ell(\mu_j^2+\delta)}\right)$$

$$=\frac{1}{i}\log\frac{b^2m^2-[i\ell(\mu_j^2+\delta)-\frac{c_3m^2}{2L}]^2}{c_2^2m^2-[i\ell(\mu_j^2+\delta)+\frac{c_3m^2}{2L}]^2} \tag{B.13}$$

$$=\frac{2c_3\ell(\mu_j^2+\delta)}{c_2^2L}\left(1-\frac{1}{\frac{c_2^2}{\ell^2(\mu_j^2+\delta)^2}+1}\right)+\mathcal{O}\left(\frac{1}{L^2}\right).$$

The first part can be combined with the sum for $|m|>K$, since

$$\frac{2\ell(\mu_j\mu_{j-m}+\delta)}{L(\mu_j-\mu_{j-m})}+\frac{2\ell(\mu_j\mu_{j+m}+\delta)}{L(\mu_j-\mu_{j+m})}\simeq\frac{2c_3(\mu_j^2+\delta)}{c_2^2L}. \tag{B.14}$$

Taking the limit $K\to\infty$ (beware that $K/L\to 0$), for the second part we have

$$-\sum_{m=1}^{\infty}\frac{2c_3\ell(\mu_j^2+\delta)}{c_2^2L\left[\frac{c_2^2}{\ell^2(\mu_j^2+\delta)^2}+1\right]}=\frac{c_3\ell(\mu_j^2+\delta)}{c_2^2L}\left[1-\frac{\pi\ell(\mu_j^2+\delta)}{c_2}\coth\left(\frac{\pi\ell(\mu_j^2+\delta)}{c_3}\right)\right]. \tag{B.15}$$

Substituting back in the values in Eq. (B.10), we will obtain the finite-size correction in Eq. (3.18). In addition, the finite-size correction in terms of $\zeta$ variable takes the form

$$\frac{\pi\rho'(\zeta)\ell^2(1+\delta\zeta^2)^2}{L}\coth\left[\pi\ell(1+\delta\zeta^2)\rho(\zeta)\right]+\mathcal{O}\left(\frac{1}{L^2}\right). \tag{B.16}$$

## C Useful formulae for elliptic functions

We collect several useful functions and formulae used in the derivations in Section 5.2.

We begin by defining the elliptic integral of the first kind

$$K(k^2)=\int_0^1\frac{dx}{\sqrt{(1-x^2)(1-k^2x^2)}}. \tag{C.1}$$

The Jacobi elliptic function $sn(x,k^2)$ is defined as the inverse of the elliptic integral of the first kind,

$$w=sn(x,k^2),\quad x=\int_0^w\frac{dz}{\sqrt{(1-z^2)(1-k^2z^2)}}, \tag{C.2}$$

and, without ambiguity, we can put $sn(x,k^2)=:sn(x)$. Other types of Jacobi elliptic functions can be defined in a similar way,

$$w=cn(x,k^2),\quad x=\int_w^1\frac{dz}{\sqrt{(1-z^2)(1-k^2+k^2z^2)}}, \tag{C.3}$$

and

$$w=dn(x,k^2),\quad x=\int_w^1\frac{dz}{\sqrt{(1-z^2)(z^2+k^2-1)}}, \tag{C.4}$$

such that

$$\mathrm{sn}^2 x + \mathrm{cn}^2 x = 1, \quad \mathrm{k}^2 \mathrm{sn}^2 x + \mathrm{dn}^2 x = 1. \tag{C.5}$$

When shifting the argument by one quarter of the period of $K(\mathrm{k}^2)$, we have

$$\mathrm{cn}\left(x + K(\mathrm{k}^2)\right) = -\sqrt{1-\mathrm{k}^2}\,\frac{\mathrm{sn}(x)}{\mathrm{dn}(x)}, \quad \mathrm{cn}\left(x + K(\mathrm{k}^2)\right) = \sqrt{1-\mathrm{k}^2}\,\frac{1}{\mathrm{dn}(x)}. \tag{C.6}$$

In addition, we also make use of theta functions to express the spin field. The most important one here is

$$\vartheta_3(z, \tau) = \sum_{n=-\infty}^{+\infty} e^{i\pi\tau n^2 + 2iz n}. \tag{C.7}$$

For a more detailed exposition and other properties of elliptic functions we refer the reader to Refs. [89, 90].

# D Numerical tests

## D.1 Gaudin norm

We present the data for several numerical checks. Firstly, we computed the Gaudin norm of a one-cut solution without a condensate. Secondly, we include a condensate, and consider two regimes: (i) isotropic interaction with $\delta = 0$ and (a) anisotropic regimes with $\delta > 0$. Case (i) without a condensate has been studied in Ref. [31], and we use it as a benchmark. Case (a) is more interesting, as it enables a non-trivial quantitative confirmation of our proposal for determining the location of a condensate or additional contours, cf. Sec. 6.2.1.

We next present our numerical results for the Gaudin norm computed on the one-cut solution with mode number $n = 1$ and filling fraction $\nu = 0.1$, for both cases (i) and (a). For this choice of parameters, there is no condensates involved. A linear fit on the finite-size numerical data yields

$$\log\mathcal{N} - \frac{1}{2}\log L = 0.00714654(1)\,L + 0.068763(9), \quad \delta = 0, \tag{D.1}$$

and

$$\log\mathcal{N} - \frac{1}{2}\log L = 0.0083405(3)\,L + 0.083756(1), \quad \delta = 1. \tag{D.2}$$

Comparing these results to those obtained from the functional integral approach, cf. Eq. (7.2) (denoted by $C_1$ in the table below), we have

|  | $C_1$ numerical | $C_1$ functional |
|---|---|---|
| $\delta = 0$ | 0.00714654(1) | 0.007156(1) |
| $\delta = 1$ | 0.0083405(3) | 0.008383(8) |

We can see that the functional approach (only requiring the knowledge of the density contours) yields very accurate results in both the isotropic (i) and anisotropic (a) case.

Next up, we analyse the cases (i) and (a) with an extra condensate, computing the Gaudin norm for a one-cut solution with mode number $n = 1$ and filling fraction $\nu = \frac{1}{3}$. In the functional integral approach, this amounts to compute the integral in Eq. (7.2) along a contour $\mathcal{C}$ comprising of three parts,

$$\mathcal{C} = \mathcal{C}_1 + \mathcal{C}_2 + \mathcal{C}_{\mathrm{cond}}. \tag{D.3}$$

Here $\mathcal{C}_1$ pertains to the original contour with density $\rho_1(\zeta)$ in Eq. (6.7) (green dashed line in panel (b) in Fig. 7), whereas contour $\mathcal{C}_2$ has density $\rho_2(\zeta)$, depicted in Eq. (6.7) by yellow

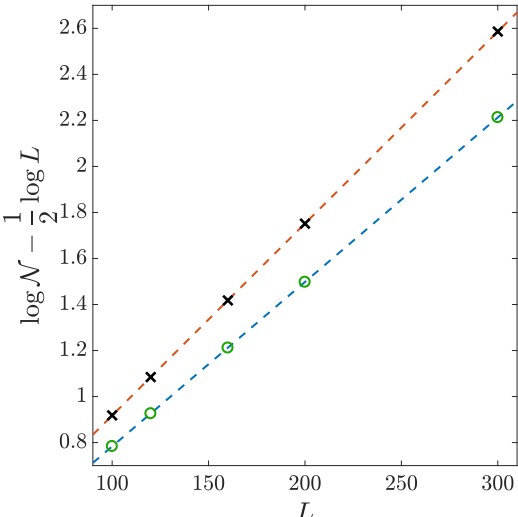

Figure 12: Logarithm of the Gaudin norm, shown for the one-cut solution with $n = 1$ and $\nu = 0.1$. Green circles (black crosses) show numerical results for $\delta = 0$ ($\delta = 1$), respectively. Linear fits are indicated by dashed lines.

dashed line in panel (b) of Fig. 7. Finally, $\mathcal{C}_{\mathrm{cond}}$ is the straight condensate contour with density $\rho_{cond} = \frac{i}{1+\delta\zeta^2}$.

This time, a linear extrapolation of the numerical finite-size data yields

$$\log\mathcal{N} - \frac{1}{2}\log L = 0.091273(6)\,L + 1.92813(4), \quad \delta = 0, \tag{D.4}$$

and

$$\log\mathcal{N} - \frac{1}{2}\log L = 0.082597(1)L + 2.09809(5), \quad \delta = 1, \tag{D.5}$$

while comparing to the results of the functional integral approach, see Eq. (7.2) ($C_1$ in the table below), we obtain

|  | $C_1$ numerical | $C_1$ functional |
|---|---|---|
| $\delta = 0$ | 0.091273(6) | 0.091121(9) |
| $\delta = 1$ | 0.082597(1) | 0.081761(2) |

In spite of an extra condensate, the functional integral method yields very accurate results. Even more importantly, this check provides a robust confirmation for the additional condensate contour(s). We note that any different contour, e.g. the usual arc-shaped contour without a condensate, produces an appreciable numerical mismatch.

## D.2 Slavnov overlap

In this section, we present a numerical check of an overlap formula. Here we compute the overlap between a semi-classical Bethe eigenstate with a single cut and a vacuum descendant state, which is a "domain-wall state" of the form $|\downarrow \cdots \downarrow\uparrow \cdots \uparrow\rangle$. Again, we perform computations for both the isotropic case (i) at $\delta = 0$ and for the anisotropic interaction (a) by setting the anisotropy parameter to $\delta = 1$.

The (unnormalised) overlap can be obtained from the general Algebraic Bethe ansatz determinant formula due to Slavnov with help of L'Hôpital rule (presented previously in e.g.

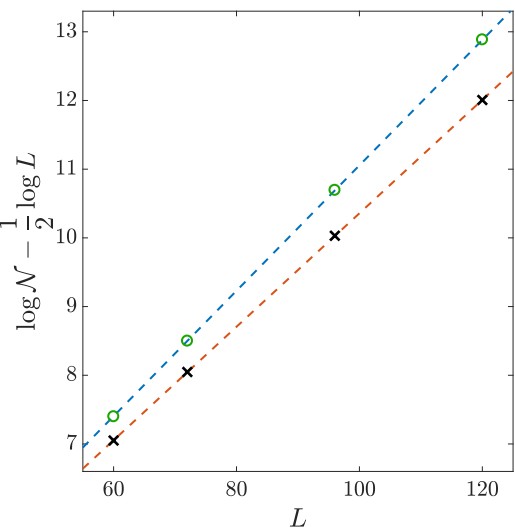

Figure 13: Logarithm of the Gaudin norm, shown for the one-cut solution with $n = 1$, $\nu = \frac{1}{3}$. Green circles (black crosses) show numerical results for $\delta = 0$ ($\delta = 1$). Linear fits are indicated by dashed lines.

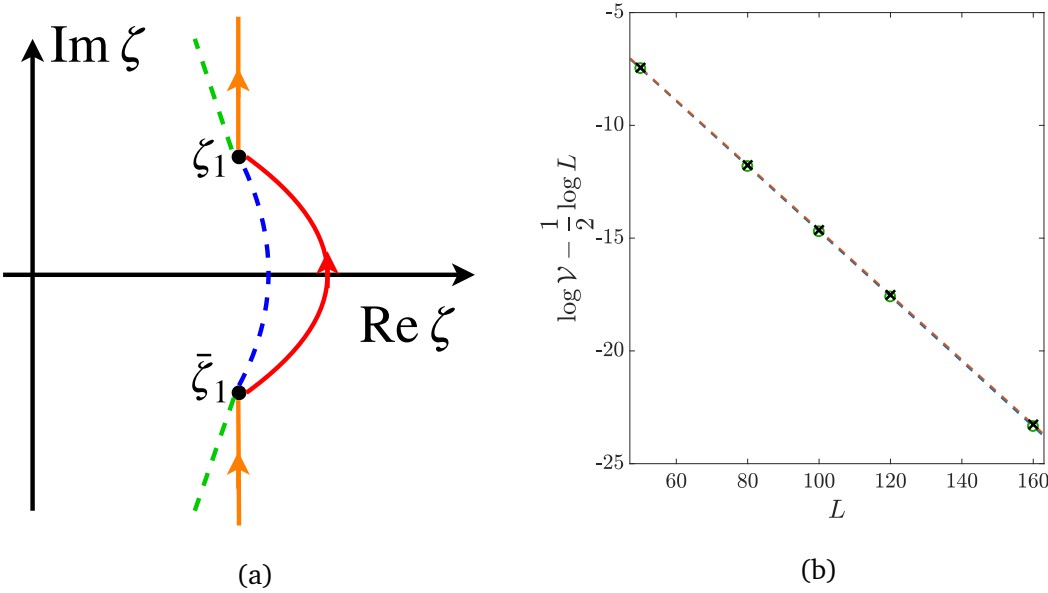

(a)

(b)

Figure 14: (a) Integration contours for computing the overlap coefficient between a one-cut state and the vacuum descendant. Square-root branch cut of $\mathfrak{p}(\zeta)$ is indicated by blue dashed line, whereas all the additional branch cuts due to the dilogarithm function are marked by green dashed line. The integration contour, marked in red (orange) on the upper (bottom) Riemann sheets, escape to infinity on the bottom sheet. (b) Logarithm of the overlap coefficient between the one-cut solution with $n = 1$ and $\nu = 0.1$ and a vacuum descendant state. Green circles (black crosses) show numerical data for interaction parameter $\delta = 0$ ($\delta = 1$), respectively, with dashed lines corresponding to linear fits.

[91]),

$$\mathcal{V} = \langle \phi | \{\vartheta\} \rangle = \prod_{l=1}^{M} \sin\left(\vartheta_l + i\frac{\eta}{2}\right)^M \frac{\det H}{\prod_{j<k} \sin(\vartheta_j - \vartheta_k)}, \tag{D.6}$$

$$H_{ab} = \cot\left(\vartheta_a - i\frac{\eta}{2}\right)^b - \cot\left(\vartheta_a + i\frac{\eta}{2}\right)^b, \tag{D.7}$$

taking the "domain-wall state" $|\phi\rangle = |\downarrow \cdots \downarrow\uparrow \cdots \uparrow\rangle$ with $M$ down-turned spins and lattice size $L$. In the isotropic limit, we obtain

$$\mathcal{V} = \langle \phi | \{\lambda\} \rangle = \prod_{l=1}^{M} \left(\lambda_l + \frac{i}{2}\right)^M \frac{\det H}{\prod_{j<k} (\lambda_j - \lambda_k)}, \tag{D.8}$$

$$H_{ab} = \left(\frac{1}{\lambda_a - i/2}\right)^b - \left(\frac{1}{\lambda_a + i/2}\right)^b. \tag{D.9}$$

In the following we shall ignore the phase and consider only the absolute value of the overlap. We expect, similarly as previously for the Gaudin norm, the following behavior at large $L$,

$$\log|\mathcal{V}| = C_2 L + \frac{1}{2}\log L + \mathcal{O}(1), \quad C_2 \in \mathcal{O}(1). \tag{D.10}$$

The results of computations are shown in Fig. 14. By numerically fitting the finite-size data, we obtained

$$\log|\mathcal{V}| - \frac{1}{2}\log L = -0.144278(5)L - 0.272812(7), \quad \delta = 0, \tag{D.11}$$

and

$$\log|\mathcal{V}| - \frac{1}{2}\log L = -0.143827(7)L - 0.254592(6), \quad \delta = 1. \tag{D.12}$$

Before we can repeat the computation using Eq. (7.6), we need to find the "quasi-momentum" corresponding to the vacuum descendant $|\phi\rangle$. With aid of Algebraic Bethe ansatz, a vacuum descendant state (with no inhomogeneities) is given by [91]

$$|\phi\rangle = |\underbrace{\downarrow \cdots \downarrow}_{M}\uparrow \cdots \uparrow\rangle = \lim_{\xi_j \to 0} \prod_{j=1}^{M} B(\xi_j)|0\rangle, \quad |0\rangle = |\uparrow \cdots \uparrow\rangle, \tag{D.13}$$

where $|0\rangle$ is the ferromagnetic Bethe vacuum $|\uparrow \cdots \uparrow\rangle$, and $B(\lambda)$ is the magnon excitation operator corresponding to the upper off-diagonal element of the quantum monodromy matrix.

The density of the "off-shell Bethe roots" can then be expressed as

$$\rho_\phi(\zeta) = \nu_1 \delta(\zeta), \quad \xi_1, \cdots \xi_M \to 0. \tag{D.14}$$

With no loss of generality, we set subsequently the classical period to $\ell = 1$. The quasi-momentum associated to $|\phi\rangle$ then reads

$$\mathfrak{p}_\phi(\zeta) = \ell \int_{\mathcal{C}_\phi} d\zeta' \frac{\rho(\zeta')(1 + \delta\zeta\zeta')}{\zeta - \zeta'} - \frac{\ell}{2\zeta} = \frac{\nu_1 - 1/2}{\zeta}. \tag{D.15}$$

Now we are ready to employ the functional integral formula (7.6). The appropriate choice of contours is shown in panel (a) in Fig. 14 [13]. Comparing the two computations of coefficient $C_2$, we find

---

[13]The rationale behind this choice is to avoid the branch cuts of the dilogarithm function. More details on this can be found in [33].

|  | $C_2$ numerical | $C_2$ functional |
|---|---|---|
| $\delta = 0$ | -0.144278(5) | -0.144485(3) |
| $\delta = 1$ | -0.143827(7) | -0.142267(2) |

Once again the computation using formula (7.6) works quite well, both in the isotropic and anisotropic cases.

# E  Numerical recipes

## E.1  Numerical solution to Bethe equations

We outline how to numerically solve for the Bethe roots to equations (3.3) (for finite but possibly large system length $L$) for a specific class of quantum eigenstates that in the thermodynamic limit become one-cut classical solution. To this end, we employ the algorithm described in Section 7 of Ref. [13] for the rational Bethe equations (i.e. for isotropic interaction, $\delta = 0$).

We use this method in combination with another method, given in Appendix C of Ref. [71], where the solution to the isotropic chain is used as the initial condition for the Newton-Raphson iteration during which the anisotropy parameter gets gradually increased. However, while this procedure works quite well for the simplest case of mode number $n = 1$, we could not achieve good convergence for mode numbers $n > 2$ and consequently could not perform any benchmark on classical solutions with two or more cuts.

## E.2  Determining branch points from filling fractions and mode numbers

We describe a numerical procedure to determine the branch points from a given set of moduli, that is the mode numbers and filling fractions. The method is completely general and applies to solutions with an arbitrary number of cuts. For simplicity however, we demonstrate it below on the class of two-cut solutions.

There are four branch points $\{\mu_1, \bar{\mu}_1, \mu_2, \bar{\mu}_2\}$ that appear in complex-conjugate pairs. Thus, there are in total four real parameters (real and imaginary components of each branch point). The finite-gap solution is parametrised by equivalently four parameters, i.e. mode numbers and filling fractions of both branch cuts, relating to the previous four parameters in a nonlinear manner. In addition, the solution must be periodic with the period $\ell$, which adds an additional constraint.

We would like to remark that, unlike in the one-cut case, the determination is highly nonlinear, related to elliptic functions and integrals. Hence, there is not a simple analytic closed-form formula available in this case. Instead, we are going to use the following numerical procedure:

- We first fix the real part of branch points of the first cut to $a \equiv \mathrm{Re}\,\mu_1 = 1$.

- We next scan a range of values $\mathrm{Im}\mu_1 \in (b_1, b_2)$ and $\mathrm{Re}\,\mu_2 \in (c_1, c_2)$, and find the value of $\mathrm{Im}\,\mu_2$ that yields the required mode numbers $n_1$ and $n_2$. More specifically, for any $\mathrm{Im}\,\mu_2$, we compute $\frac{d\mathfrak{p}}{\ell}$ by demanding the $\mathcal{A}$-cycle to vanish. Since the classical period reads

$$\ell = \frac{2\pi n_1}{\int_{\mathcal{B}_1} d\mathfrak{p}/\ell}, \tag{E.1}$$

we can numerically determine $\mathrm{Im}\,\mu_2$ from the requirement

$$\int_{\mathcal{B}_2} d\mathfrak{p} = 2\pi n_2. \tag{E.2}$$

- Having done the above, we can readily compute following quantities,

$$\tilde{\nu}_1 = \oint_{\mathcal{A}_1} \frac{\mathrm{d}\mathfrak{p}}{\mu}, \quad \tilde{\nu}_2 = \oint_{\mathcal{A}_2} \frac{\mathrm{d}\mathfrak{p}}{\mu}, \tag{E.3}$$

which moreover depend on $\mathrm{Im}\,\mu_1$ and $\mathrm{Re}\,\mu_2$.

- By requiring that $\tilde{\nu}_1 = \nu_1$, we obtain a "curve" in the plane spanned by $\mathrm{Im}\mu_1$ and $\mathrm{Re}\,\mu_2$. By finally requesting also that $\tilde{\nu}_1 = \nu_2$, we are left with a single point, say $(b, c)$. The last point, call it $d$, corresponds to $\mathrm{Im}\,\mu_2$.

- We have thus determined to complex branch points $\mu_1 = 1 + bi$, $\mu_2 = c + di$ (alongside their complex conjugates) which yields the prescribed classical period $\ell$, mode number $n_1$, $n_2$ and filling fraction $\nu_1$, $\nu_2$ of a general two-solution.

In making a comparison with the Bethe root distributions, we normally prefer to set the classical period to $\ell = 1$. In this case we simply divide the above branch points by $\ell$, see Eq. (E.1), that is

$$\mu_{1,n} = \frac{1}{\ell} + \frac{b}{\ell} i, \quad \mu_{2,n} = \frac{c}{\ell} + \frac{d}{\ell} i, \tag{E.4}$$

or equivalently in terms of spectral parameter $\zeta = 1/\mu$,

$$\zeta_{1,n} = \frac{1}{\bar{\mu}_{1,n}}, \quad \zeta_{2,n} = \frac{1}{\bar{\mu}_{2,n}}. \tag{E.5}$$

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
