# Peer review of "Semi-classical quantisation of magnetic solitons in the anisotropic Heisenberg quantum chain"

_SciPost Physics, doi:SciPost Phys. 10, 086 (2021)_

## Round 4 · Referee Report · Anonymous (Referee 1) · 2020-12-4

Strengths

Highly detailed
Pedagogically written
Carefully written
Important results

Weaknesses

Originality

Report

The paper gives a semiclassical treatment in the anisotropic Heisenberg quantum chain. The paper incorporates the most important methods to give semiclassical results in exactly solvable system, by using namely the algebro-geometrical approach, semi-classical quantization, and an astute use of determinantal formulas. All these methods are carefully woven together to give a rather comprehensive treatment of this model in the semiclassical limit. The paper also contains a good introduction based on the harmonic oscillator, which is also very useful. As a whole I think the paper will be an invaluble reference in its field. I recommend the paper to be published in SciPost.

---

## Round 4 · Referee Report · Anonymous (Referee 2) · 2020-12-29

Strengths

  1. Thorough analysis supported by numerics
  2. Complete classification of semi-classical states of the XXZ spin chain
  3. Many examples considered in detail

Weaknesses

  1. none

Report

The authors study semi-classical states of the Heisenberg model using Bethe ansatz and the finite-gap integration method. Their analysis is very thorough and comprehensive. Along with the general formalism a few examples are studied at length, such as macroscopic magnetic waves and the bion solution.

The paper can be published as is, but I believe it will benefit from a few minor improvements.

Requested changes

  1. What determines the spacial period of the solution? From the discussion on p.20 and eqs. 4.43, 4.44 it seems to be the asymptotics of the quasi-momentum at infinity. This is indeed the case for the 1-cut solution, but for the 2-cut case the period depends on moduli. Why?

  2. The authors call $n_j$ winding numbers and mode intermittently. These are physically distinct notions. The term better reflecting the physics should be used uniformly.

  3. Deviations of the $\sigma^z \sigma^z$ correlator in fig. 11b from 8.5 are way too large to be accounted for by finite-size effects, if estimated from $\sigma^x \sigma^x$ where the agreement is almost perfect. This difference needs to be explained, or else the authors should provide a reliable estimate of finite-size corrections. Otherwise the discrepancy casts doubts on the conjectural relation between the phase average and quantum correlators.

  4. Limits of integration in 2.38 and 2.41 are inverted.

  5. I'd suggest to proofread the text English-wise.

  • validity: high
  • significance: top
  • originality: good
  • clarity: high
  • formatting: good
  • grammar: below threshold

---

## Round 5 · Referee Report · Anonymous (Referee 2) · 2021-3-22

Report

Can be published.

Requested changes

No further ones

---

## Round 5 · Author Response

We are grateful to the Referees for their assessment of our manuscript and for their valuable comments and suggestions. Our replies to the queries are given below.

  1. What determines the spacial period of the solution? From the discussion on p.20 and eqs. 4.43, 4.44 it seems to be the asymptotics of the quasi-momentum at infinity. This is indeed the case for the 1-cut solution, but for the 2-cut case the period depends on moduli. Why?

Period $\ell$ prescribes the circumference of the classical phase space. It enters simply as the overall scale in the quasi-momentum. Wavenumbers of nonlinear finite-gap modes are given by $k_i = (2\pi/\ell)n_i$ for integer mode numbers $n_i$. In the one-cut case, we can fix the value of \ell. By choosing a mode number n we can determine the single wavenumber k. In the two-cut case, general quasi-periodic solutions depend on two complex branch points (four real parameters). By setting the period $\ell$ fixed, this does not yield periodic solutions in general. Those are only a subset of solutions for which wavenumbers are integer multiples of $2\pi / \ell$. In our case, the algebraic data can be read off from the coefficients of the polynomial $\mathcal{R}(\lambda)$ (see Eq. (4.15)). The spatial periods are defined in Eqs. (4.29) and (4.33) via Abelian integrals. The quasi-momentum is also defined from the same algebraic data. Its expansion at infinity generates values for the conserved quantities of the given finite-gap solution. Given the mode numbers (together with their partial filling fractions), there is no algebraic procedure to compute the associated branch points for cases with more than 2 cuts. Indeed, ensuring integrality of the B-cycles may be viewed as the classical version of the Bethe equations. In practice we employ ``reverse procedure'' to find the branch points for 2-cut solutions, as described in Appendix E.2.

  1. The authors call $n_j$ winding numbers and mode intermittently. These are physically distinct notions. The term better reflecting the physics should be used uniformly.

We followed the advice and uniformly replaced "winding numbers" by "mode numbers".

  1. Deviations of the $\sigma_z \sigma_z$ correlator in fig. 11b from 8.5 are way too large to be accounted for by finite-size effects, if estimated from $\sigma_x \sigma_x$ where the agreement is almost perfect. This difference needs to be explained, or else the authors should provide a reliable estimate of finite-size corrections. Otherwise the discrepancy casts doubts on the conjectural relation between the phase average and quantum correlators.

Unfortunately, we presently only have access to the correlation functions of eigenstates with at most M=6 Bethe roots (for the 1/3 filling fraction). This is arguably "far away" from an approximate semi-classical regime. While the mismatch in the longitudinal correlators is admittedly quite large, the observed matching in the transversal sector suggests that our conjectural prescription is accurate. In our opinion, guided also by Smirnov's results on the form-factors, we are eventually looking at a pronounced finite-size effect. We do not know of any better computational scheme to further improve our numerical results and study the convergence. We cannot think of any feasible (let alone reliable) finite-size analysis for such small system sizes in the absence of any natural expansion parameter. Our hope is that this section of the paper, despite being conjectural, can at least stimulate further research on this aspect. In the revised version we have entirely rewritten Section 8 to emphasise clearly what has been computed numerically.

  1. Limits of integration in 2.38 and 2.41 are inverted.

We thank the referee for spotting this. We have made appropriate adjustments.

  1. I'd suggest to proofread the text English-wise.

In the revised version we have substantially improved the text and resolved grammatical issues.

---

## Round 5 · List of Changes

List of changes:

  1. We have corrected several typographical and grammatical mistakes/errors of the article. We have redrafted Section 8 as suggested by the referee.

  2. We have changed the use of winding number to mode numbers.

  3. For Eq. (2.38) and (2.41), we change two limits in the equations to a single limit.

  4. We correct a sign typo in Eq. (3.15), (3.16) and (3.21).

---

## Editorial Decision

published